# LoD-Loc: Aerial Visual Localization using LoD 3D Map with Neural Wireframe Alignment

**Juelin Zhu**[1*]
zhujuelin@nudt.edu.cn

**Shen Yan**[1*]
yanshen12@nudt.edu.cn

**Long Wang**[2]
wanglongzju@gmail.com

**Shengyue Zhang**[1]
zhangshengyue23@nudt.edu.cn

**Yu Liu**[1]
jasonyuliu@nudt.edu.cn

**Maojun Zhang**[1†]
mjzhang@nudt.edu.cn

[1]National University of Defense Technology    [2]SenseTime Research

## Abstract

We propose a new method named LoD-Loc for visual localization in the air. Unlike existing localization algorithms, LoD-Loc does not rely on complex 3D representations and can estimate the pose of an Unmanned Aerial Vehicle (UAV) using a Level-of-Detail (LoD) 3D map. LoD-Loc mainly achieves this goal by aligning the wireframe derived from the LoD projected model with that predicted by the neural network. Specifically, given a coarse pose provided by the UAV sensor, LoD-Loc hierarchically builds a cost volume for uniformly sampled pose hypotheses to describe pose probability distribution and select a pose with maximum probability. Each cost within this volume measures the degree of line alignment between projected and predicted wireframes. LoD-Loc also devises a 6-DoF pose optimization algorithm to refine the previous result with a differentiable Gaussian-Newton method. As no public dataset exists for the studied problem, we collect two datasets with map levels of LoD3.0 and LoD2.0, along with real RGB queries and ground-truth pose annotations. We benchmark our method and demonstrate that LoD-Loc achieves excellent performance, even surpassing current state-of-the-art methods that use textured 3D models for localization. The code and dataset are available at `https://victorzoo.github.io/LoD-Loc.github.io/`.

## 1 Introduction

Aerial visual localization is the process of determining the global position and orientation for a UAV camera relative to a known map. This process benefits many important applications, ranging from cargo transport [69], surveillance [19, 72], to search and rescue [15, 63].

Following localization algorithms on the ground [18, 30, 42, 53, 54, 57, 67, 68, 70, 77], current aerial visual localization approaches [22, 72] typically involve matching pixels in a query image with points in a pre-built high-quality 3D map, which is often derived from 3D texture models [43, 76, 72]. Subsequently, a Perspective-n-Point (PnP) RANSAC [32, 37, 14, 23, 27, 24] technique is commonly used to calculate the camera pose. However, building high-quality 3D maps using photogrammetry [12, 28, 64, 58, 33] is expensive on a global scale and requires frequent updates to account for temporal changes in visual appearance. Besides, these 3D maps are costly to store, which poses significant challenges for terminal deployment on drones. Furthermore, high-resolution 3D maps disclose detailed information about the localization area, raising critical concerns regarding homeland security and privacy preservation.

---

[*]Equal contribution
[†]Corresponding author

38th Conference on Neural Information Processing Systems (NeurIPS 2024).

6-DoF Pose Estimation over **LoD Model**

☑ Easy Acquire/Maintain
☑ Light-weight Map Size
☑ Privacy Preservation

LoD-Loc

Sensor Prior          Localization Result

Figure 1: In this paper, we propose LoD-Loc to tackle visual localization w.r.t a scene represented by a LoD 3D map, characterized by its ease of acquisition, lightweight nature, and built-in privacy-preserving capabilities. Given a query image and its coarse sensor pose, our method utilizes the wireframe alignment of LoD models to recover the camera pose.

To address these challenges, we propose to leverage the Level of Detail (LoD) 3D city maps [38, 16] as the cue for localization, as illustrated in Figure 1. Compared with traditional textured 3D models, LoD 3D models enjoy the following advantages: 1) **Ease of acquisition and maintenance**: World-scale LoD city models can be generated with the rapid development of remote sensing [17, 34, 48, 52, 31]. Many commercial companies, such as Google Maps [8] and Baidu Maps [7], have already integrated LoD 3D models into their MAP Applications. 2) **Light-weight map size**: LoD maps are extremely compact, up to $10^4$ times smaller in size than textured 3D maps, enabling on-device localization over large areas. 3) **Privacy preservation and policy-friendly**: As LoD city models only reveal the basic 3D outlines of buildings in a highly abstracted and simplified manner, they raise fewer concerns about the disclosure of privacy and land resource secrets.

However, compared with a textured 3D model, using LoD maps for localization is very challenging, primarily due to the lack of texture and detail. This deficiency makes it nearly impossible to establish local feature-based 2D-to-3D correspondences. Inspired by the idea that, when the pose is correctly solved, the network-predicted building wireframes can align with those projected from the LoD 3D model, as shown in Figure 1, we introduce LoD-Loc, a novel approach tailored for visual localization in LoD 3D city maps. Our method takes a query image and its real sensor data (i.e., GPS, gravity, and compass) as input, and estimates the 6-DoF pose of a drone in flight. Specifically, we initially fix the 2-DoF gravity direction and generate pose hypotheses by sampling 4-DoF (comprising position and yaw angle) around the sensor pose, given that the gravity direction provided by the inertial unit exhibits minor error. Following the generation of pose hypotheses, LoD building wireframes are projected onto the query image plane. Each pose hypothesis is then scored based on the alignment between the projected and predicted wireframes, thereby forming a 4D pose cost volume. By applying a *softmax* operation, we derive a probability density over the pose, which can be used for pose selection through classification. Moreover, after the pose selection stage, a differentiable Gauss-Newton method, with an optimization objective to maximize the wireframe alignment, is employed to refine the overall 6-DoF pose. The pose selection and optimization processes are fully differentiable w.r.t. the network output, which enables the use of ground-truth poses as supervision for training feature extraction and pose estimation in an end-to-end manner.

To achieve high accuracy and low memory usage, we propose a hierarchical scheme for pose selection that utilizes multiple small pose volumes, to progressively compute poses in a coarse-to-fine manner. Throughout the hierarchy, we adopt an adaptive sampling strategy, where the variance-based uncertainty from the previous stage influences the sampling range of the next stage for constructing pose cost volume. This adaptive process enables reasonable and fine-grained spatial partitioning of poses, resulting in a significant improvement in the final pose output.

To facilitate research in this area, and to train and evaluate our method, we release two datasets with map levels of LoD3.0 and LoD2.0, respectively, as shown in Figure 2. For the LoD3.0 dataset, we employ a semi-automatic method to generate LoD model data from a recent large-scale oblique photography scene [72], covering an area of 2.5 square kilometers. The query images are captured by drones, with sensor data (e.g., GPS, IMU) recorded. For the LOD2.0 dataset, we use LoD model data provided by the Swiss federal authorities, specifically the SwissTOPO [9–11] data near École

Table 1: **Differenet types of maps for visual localization.**

| Map Type | SfM SLAM | Mesh model | Satellite images | OpenStreetMap | LoD model (**our works**) |
|---|---|---|---|---|---|
| What? | 3D points +features | textured meshes | pixel intensity | polygons, lines, points | wireframes, faces |
| Explicit geometry? | 3D | 3D | × | 2D | 3D |
| Visual appearance? | ✓ | ✓ | ✓ | × | × |
| x-DoF pose estimation | 6-DoF | 6-DoF | 3-DoF | 3-DoF | 6-DoF |
| Storage per 1km$^2$ | 42 GB | 9.8 GB | 75 MB | 4.8 MB | 2.84 MB |
| Size reduction v.s. SfM | - | 4.28× | 550× | 8800× | 15100× |

Polytechnique Fédérale de Lausanne (EPFL), covering an area of 8.2 square kilometers. The query images with ground-truth poses are sourced from the CrossLoc [74] project.

We conduct extensive experiments on these two datasets. The results show that, due to the lack of both color and texture in the LoD 3D model, previous state-of-the-art image retrieval-and-matching methods [44, 53, 54, 29, 41, 25, 66, 46, 47] basically fail. In contrast, our method consistently achieves excellent results, even surpassing current state-of-the-art methods [53, 54, 66, 72] that use textured 3D models for localization.

**Contributions.**

- We propose the use of Level of Detail (LoD) 3D maps for 6-DoF visual localization in the air.
- We introduce a novel localization method that utilizes wireframe alignment for pose estimation.
- Our method is differentiable, allowing the pipeline to be trained end-to-end with pose supervision.
- We release two LoD city datasets, complete with RGB queries and ground-truth pose annotations.

## 2 Related Works

**Localization from SfM or Mesh Map.** SfM maps typically consist of reference images and 3D track points with their associated features [58]. For a given query image, an image retrieval method [13, 29] is initially utilized to identify co-visible reference images. Following this, feature matching algorithms [41, 25, 54, 66] are employed to establish accurate 2D-2D correspondences between the query image and the identified reference images, with track information being used to transform these 2D-2D correspondences into 2D-3D relationships. Finally, the pose is resolved using PnP RANSAC [32, 37, 14, 23, 27, 24, 77].

Mesh maps are typically defined by a textured mesh model. Initially, reference images with depth are rendered at appropriate viewpoints surrounding the model [72, 43, 44, 76]. Similar image retrieval [13, 29] and matching [41, 25, 54, 66] processes are employed to identify co-visible images and to establish 2D-2D correspondences. The depth map is utilized to transform 2D-2D correspondences into 2D-3D relationships, and the pose is subsequently determined by a PnP RANSAC [32, 37, 14, 23, 27, 24, 77].

Despite providing high-accuracy localization results, both SfM and mesh models present significant challenges in terms of reconstruction and maintenance. Additionally, their extensive size complicates deployment, necessitating its existence solely in the cloud. Moreover, these maps raise serious concerns about the privacy of personal and land resource leakage.

**Localization from other types of Map.** To mitigate these issues, researchers have proposed to use alternative types of maps for localization. In addressing the difficulty of map reconstruction and maintenance, some methods opt for overhead imagery such as satellite [59, 60, 73, 56], or leverage OpenStreetMap [55] as a reference. However, these methods are limited to estimating a 3-DoF (planar position and heading) pose at most. To address the issue of large map size, some methods have made attempts to compress the maps [20, 21, 78], reduce model complexity [44], or utilize geometry information without features [49, 79, 39, 75]. In the context of privacy, some methods [36, 61, 65] propose to transform 3D point clouds into 3D line clouds, leverage semantic

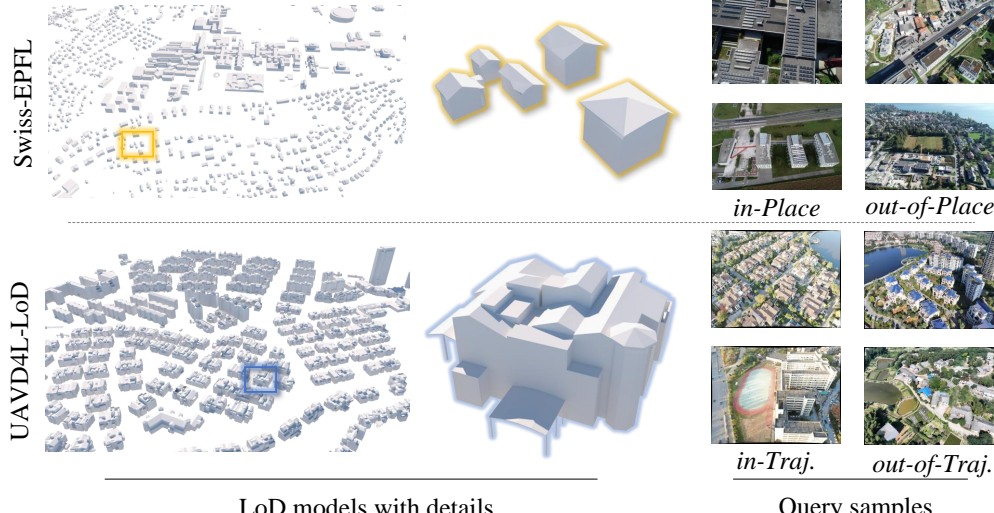

Figure 2: **Overview of datasets.** The left side shows the LoD models of the released data. The LoD2.0 model from Swiss-EPFL includes building height and roof information, while the LoD3.0 model from UAVD4L-LoD contains more detailed structural information such as building height, roof, and side pillars. The right side illustrates samples of query images, which consist of images captured by drones in various scenes.

information from point clouds, or utilize semantic 3D maps to enhance privacy [50]. Some other methods apply learning-based pose regression or scene point regression models [35, 62, 40] that do not explicitly store the 3D map. However, the effectiveness and generalizability of these methods are often inferior to those that rely on SfM or texture mesh maps. A detailed comparison of attributes from different maps is provided in Table 1.

## 3 Method

Given a 3D city LoD map $\mathcal{M}$, a query image $\mathbf{I}$, and its coarse sensor pose $\boldsymbol{\xi}_p$, the goal of the proposed method is to compute the absolute 6-DoF pose $\boldsymbol{\xi}^*$. First, a convolutional neural network is used to extract the wireframe probability map for the query image $\mathbf{I}$ at multiple levels (Sec. 3.1). Second, at each level, uniform pose sampling and 3D wireframe projection are employed to build a cost volume for various pose hypotheses, describing the pose probability distribution. The pose with the maximum probability is then selected (Sec. 3.2). Finally, a post-processing network refines the wireframe probability map after the last level, and a Gauss-Newton method is applied to refine the pose chosen in the previous stage (Sec. 3.3). Figure 3 provides an overview of the proposed method.

### 3.1 Multi-Scale Feature Extractor

We use a standard convolutional architecture with U-Net [51] to extract multi-level features from the query image $\mathbf{I}$. Different from previous works that maintain a high-dimensional feature map to encapsulate rich visual information for each level, we abstract and reduce the feature map dimension to a single channel, where each pixel in this map signifies the likelihood of being a wireframe. The resulting feature maps are denoted by $\mathbf{F}_l \in \mathbb{R}^{H_l \times W_l \times 1}$, where $l = \{1, 2, 3\}$ is the level index. More details on the architecture of the proposed network can be found in Appendix D.1.

### 3.2 Pose Selection from Cost Volume

After feature extraction, we construct a cost volume based on various pose hypotheses sampled around the coarse sensor pose, selecting the pose with the highest probability at each level. To ensure

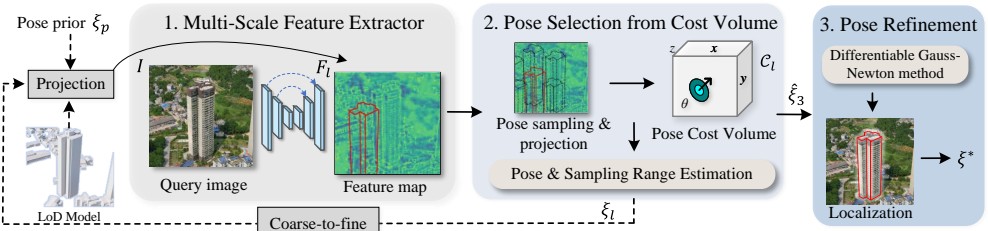

Figure 3: **Overview of LoD-Loc**. 1. LoD-Loc employs a CNN to extract multi-level features $\mathbf{F}_l$ for the query image $\mathbf{I}$ (Sec. 3.1). 2. A cost volume $\mathcal{C}_l$ is built for various pose hypotheses sampled around the coarse sensor pose $\boldsymbol{\xi}_p$ to select the pose $\boldsymbol{\xi}_l$ with the highest probability, based on the projected wireframe of the 3D LoD model (Sec. 3.2). 3. A differentiable Gauss-Newton method is used to refine the final selected pose $\boldsymbol{\xi}_3$, to obtain a more accurate pose $\boldsymbol{\xi}^*$ (Sec. 3.3).

efficient sampling, we utilize the uncertainty in pose selection at the current level to determine the pose sampling range for the next level.

**Pose cost volume reconstruction.** This subsection explains how to construct the pose cost volume. For a specific level $l$, the initial pose is represented by $\boldsymbol{\xi}_l$, and the computed pose is denoted as $\widehat{\boldsymbol{\xi}}_l$. As part of a progressive process, we keep $\boldsymbol{\xi}_1 = \boldsymbol{\xi}_p$ and $\boldsymbol{\xi}_{l+1} = \widehat{\boldsymbol{\xi}}_l$. The pose $\boldsymbol{\xi}_l$ can be decoupled into six degrees, with $\boldsymbol{\xi}_l = (\mathrm{x}_l, \mathrm{y}_l, \mathrm{z}_l, \theta_l, \varphi_l, \psi_l)$, where $(\mathrm{x}_l, \mathrm{y}_l, \mathrm{z}_l)$ represents the translation in 3D space while $(\theta_l, \varphi_l, \psi_l)$ refers to the Eular angles (i.e., yaw, pitch, roll). Since the pitch and roll $(\varphi_p, \psi_p)$ of the gravity direction from the sensor pose data exhibit high accuracy, we fix $(\varphi_l, \psi_l) = (\varphi_p, \psi_p)$ and only conduct operations on the remaining 4-DoF (i.e., $(\mathrm{x}_l, \mathrm{y}_l, \mathrm{z}_l, \theta_l)$ ) in the following steps.

Specifically, we begin by uniformly sampling 4-DoF poses centered on the initial pose $\boldsymbol{\xi}_l$, with the sampling range and number defined as $\mathbf{r}_l = [\mathrm{r}_l(\mathrm{x}), \mathrm{r}_l(\mathrm{y}), \mathrm{r}_l(\mathrm{z}), \mathrm{r}_l(\theta)]$ and $[m_l(\mathrm{x}), m_l(\mathrm{y}), m_l(\mathrm{z}), m_l(\theta)]$, respectively. The pose hypothesis $\{\boldsymbol{\xi}_l^{hyp}(d)\}$ is generated along $(\mathrm{x}, \mathrm{y}, \mathrm{z}, \theta)$ directions separately, where $d \in (\mathrm{x}, \mathrm{y}, \mathrm{z}, \theta)$.

$$\{\boldsymbol{\xi}_l^{hyp}(d)\} = \{\underbrace{-\mathrm{r}_l(d)/2 + d_l, \cdots, d_l, \cdots, d_l + \mathrm{r}_l(d)/2}_{m_l(d)}\}. \tag{1}$$

Next, for a given pose hypothesis, denoted as $\boldsymbol{\xi}_l^{hyp} = (\mathbf{R}_l^{hyp}, \mathbf{t}_l^{hyp})$ and a set of discrete 3D wireframe points denoted as $\{\mathbf{P}_i\}$, we define a line alignment cost:

$$\mathcal{C}_l(\boldsymbol{\xi}_l^{hyp}) = \frac{1}{n} \cdot \sum_{i=1}^{n} \mathbf{F}_l[\mathbf{p}_i]. \tag{2}$$

In this equation, $\mathbf{p}_i = \Pi(\mathbf{R}_l^{hyp} \cdot \mathbf{P}_i + \mathbf{t}_l^{hyp})$ represents the projection of 3D point $\mathbf{P}_i$ under pose hypothesis $\boldsymbol{\xi}_l^{hyp}$ and $[\cdot]$ denotes a lookup with sub-pixel interpolation. The construction of the 3D wireframe points $\{\mathbf{P}_i\}$ is provided in the next paragraph. By combining these costs in a grid manner across four distinct dimensions $(\mathrm{x}, \mathrm{y}, \mathrm{z}, \theta)$, we obtain a pose cost volume $\mathcal{C}_l$ with dimensions $[m_l(\mathrm{x}) \times m_l(\mathrm{y}) \times m_l(\mathrm{z}) \times m_l(\theta)]$. Finally, a *softmax* function is applied to $\mathcal{C}_l$ to yield a probability distribution volume $\mathcal{P}_l$. For pose inference, we select the pose $\widehat{\boldsymbol{\xi}}_l$ with maximum probability by *argmax* operation upon $\mathcal{P}_l$.

**Discrete 3D wireframe points generation.** For a query image $\mathbf{I}$ and its associated sensor pose $\boldsymbol{\xi}_p$, we describe how to sample and identify discrete 3D wireframe points $\{\mathbf{P}_i\}$ across the entire LoD map $\mathcal{M}$. Assume the LoD map $\mathcal{M}$ is characterized by a number of faces with vertices $\mathcal{V}_j = [X_j, Y_j, Z_j]^T \in \mathbb{R}^3$. We derive each line of the LoD model as $\ell_{jk} = [(X_j, X_k), (Y_j, Y_k), (Z_j, Z_k)]$ by connecting vertices $\mathcal{V}_j$ and $\mathcal{V}_k$. To focus on distinct geometric structures such as building edges, we discard lines whose normals of their neighboring faces exhibit a significant difference, larger than $\mu = 10$ degrees. The line simplification process is facilitated with the assistance of Blender [1].

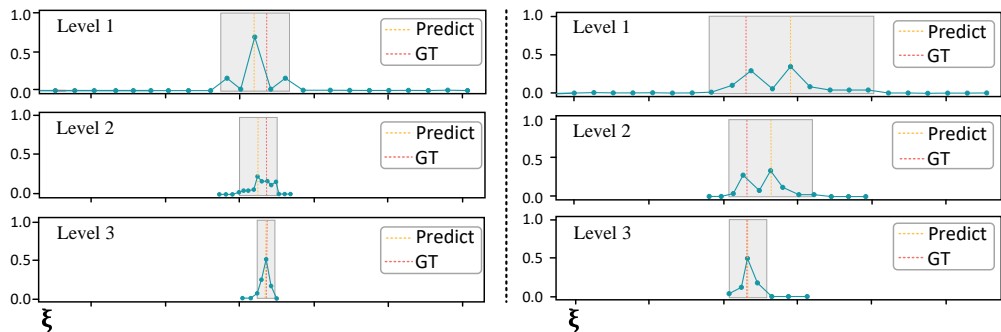

Figure 4: Toy examples to illustrate the uncertainty sampling range estimation. We show pose distribution (connected blue dots), pose prediction (yellow dash line), the ground truth pose (red dash line), and uncertainty sampling range (gray) in the three levels.

Subsequently, we set the sampling density $\delta$ in meters and uniformly sample points along all simplified lines $\{\ell_{jk}\}$ to obtain 3D points as $\{\mathbf{P}_i^u\}$, where $i$ represents the index of the 3D points. To obtain visible 3D points for a query image $\mathbf{I}$, we use the pinhole camera projection (with intrinsic matrix $\mathbf{K}^p$) and pose prior $\boldsymbol{\xi}_p = (\mathbf{R}^p, \mathbf{t}^p)$ to identify 3D points, taking factors such as frustum inside and occlusion information into account. In particular, we first project 3D points onto the 2D image plane:

$$\mathrm{d}_i[u_i, v_i, 1]^\top = \mathbf{K}^p \left( \mathbf{R}^p [X_i, Y_i, Z_i]^\top + \mathbf{t}^p \right) \tag{3}$$

where $[X_i, Y_i, Z_i] \in \{\mathbf{P}_i^u\}$, and $\mathrm{d}$ is the projected depth for point $[X_i, Y_i, Z_i]$. We then render a depth map from the LoD map $\mathcal{M}$ from pose $\boldsymbol{\xi}_p$, denoted as $\mathbf{D}$. A boolean mask is calculated as:

$$B_i = \mathrm{d}_i < \mathbf{D}(u_i, v_i) \ \& \ 0 < u_i < H \ \& \ 0 < v_i < W \tag{4}$$

where $H$ and $W$ denote the size of the image $\mathbf{I}$, $\mathbf{D}(u_i, v_i)$ means the interpolating value on depth map $\mathbf{D}$ at $(u_i, v_i)$. The final discrete visible 3D wireframe points $\{\mathbf{P}_i\}$ can be obtained using Eq. 5:

$$\{\mathbf{P}_i\} = \{\mathbf{P}_i^u[B_i]\}. \tag{5}$$

**Uncertainty sampling range estimation.** During the coarse-to-fine process, we leverage the pose selection uncertainty from the previous level to determine the sampling range of the current level. This strategy allows us to progressively subdivide the pose sampling space, thereby enhancing the precision of pose selection. More specifically, for $l = 1$, we define the pose sampling range by evaluating the error in the coarse sensor pose. The sampling range for $(\mathrm{x}, \mathrm{y}, \mathrm{z}, \theta)$ is defined as $[\mathrm{r}_1(x), \mathrm{r}_1(y), \mathrm{r}_1(z), \mathrm{r}_1(\theta)] = [\mathrm{r}_p(x), \mathrm{r}_p(y), \mathrm{r}_p(z), \mathrm{r}_p(\theta)]$. For $l = \{2, 3\}$, we employ the variance of the probability distribution volume $\mathcal{P}_{l-1}$ at $l - 1$ to determine the pose sampling range $\mathbf{r}_l$.

In particular, since the pose hypothesis $\{\boldsymbol{\xi}_l^{hyp}\}$, pose cost volume $\mathcal{C}_l$ and probability distribution volume $\mathcal{P}_l$ share the same data structure, we flatten them and index them by $t$. The variance $\mathbf{v}_l$ at level $l$ is calculated as:

$$\mathbf{v}_l = \sum_t {}^t\mathcal{P}_{l-1} \cdot \|\widehat{\boldsymbol{\xi}}_{l-1} \ominus {}^t\boldsymbol{\xi}_{l-1}^{hyp}\|^2. \tag{6}$$

Here, the symbol $\ominus$ represents the subtraction operation separately applied to the $(\mathrm{x}, \mathrm{y}, \mathrm{z}, \theta)$ directions. The corresponding standard deviation is computed as $\boldsymbol{\sigma}_l = \sqrt{\mathbf{v}_l}$. We compute the pose sampling range as $\mathbf{r}_l = 2\lambda \cdot \boldsymbol{\sigma}_l$, where $\lambda$ is a hyperparameter that adjust the length of the sample range. A visualization of this uncertainty sampling range estimation process can be found in Figure 4.

### 3.3 Pose Refinement

Based on the selected pose $\widehat{\boldsymbol{\xi}}_3$ from the previous stage, we use a refined wireframe probability map $\mathbf{F}_{rf}$, which is further extracted from the feature map $\mathbf{F}_3$ via a post-processing convolutional network, we optimize the pose $\boldsymbol{\xi}^* = (\mathbf{R}^*, \mathbf{t}^*)$ so as to align the 3D wireframe with the 2D predicted wireframe. Specifically, we define the objective function:

$$E(\boldsymbol{\xi}^*) = -\sum_i \|f_i\|^2 = -\sum_i \|\mathbf{F}_{rf}[\Pi(\mathbf{R}^* \cdot \mathbf{P}_i + \mathbf{t}^*)]\|^2, \tag{7}$$

where $\mathbf{P}_i$ is the 3D wireframe point and $\Pi$ is the projection operation. Minimizing this function aligns the projected 3D wireframe points with the 2D locations that have a higher predicted probability. The pose update formula for $\boldsymbol{\xi}^*$, as derived from the Gauss-Newton method, is given by:

$$
\begin{aligned}
\Delta\boldsymbol{\xi} &= -\sum_i (J_i^T J_i)^{-1} \sum_i (J_i^T f_i) \\
\mathbf{t}^* &= \mathbf{R}^* \cdot \Delta\boldsymbol{\xi}_t + \mathbf{t}^* \\
\mathbf{R}^* &= \mathbf{R}^* \cdot \exp(\Delta\boldsymbol{\xi}_r).
\end{aligned}
\tag{8}
$$

In this formula, $\Delta\boldsymbol{\xi} \in \mathbb{R}^6$ represents a six-dimensional transformation vector, where $\Delta\boldsymbol{\xi}_r \in \mathbb{R}^3$ constitutes the rotational component and $\Delta\boldsymbol{\xi}_t \in \mathbb{R}^3$ represents the translational component. We transform the rotational component $\Delta\boldsymbol{\xi}_r$ into a $3\times3$ rotation matrix by the exponential map of the Lie algebra $\mathfrak{so}(3)$. Besides, $J_i$ represents the Jacobian matrix of the residual function $f_i$ with respect to the pose parameters. A comprehensive explanation and detailed implementation of the Jacobian computation are provided in Appendix D.2.

### 3.4 Supervision

We employ two separate loss functions to facilitate end-to-end training of the pose selection procedure (Sec. 3.2) and the pose refinement module (Sec. 3.3). For the pose selection module, we minimize the negative log-likelihood loss on the probability distribution volume $\mathcal{P}_l$ at three levels, where $l \in \{1, 2, 3\}$.

$$
L_s = -\sum_l \log \mathcal{P}_l[\bar{\boldsymbol{\xi}}],
\tag{9}
$$

For the pose refinement process, the training involves minimizing the reprojection errors between 3D wireframe points transformed by the estimated pose $\boldsymbol{\xi}^*$ and the ground truth pose $\bar{\boldsymbol{\xi}} = (\bar{\mathbf{R}}, \bar{\mathbf{t}})$:

$$
L_f = \sum_i \rho(||\Pi(\mathbf{R}^* \cdot \mathbf{P}_i + \mathbf{t}^*) - \Pi(\bar{\mathbf{R}} \cdot \mathbf{P}_i + \bar{\mathbf{t}})||^2),
\tag{10}
$$

where $\rho$ represents the Huber robust kernel.

## 4 Experiment

Extensive experiments are conducted on the UAVD4L-LoD and Swiss-EPFL datasets to demonstrate the effectiveness of our proposed model as described in Sec. 4.2. Additionally, ablation studies are conducted on the UAVD4L-LoD dataset in Sec. 4.3.

**Datasets.** The released datasets consist of two distinct parts, named UAVD4L-LoD and Swiss-EPFL, providing LoD3.0 and LoD2.0 models, respectively. The UAVD4L-LoD dataset, which spans an area of 2.5 square kilometers, is generated through a semi-automatic process which produces a 3D LoD map from the mesh model of the UAVD4L [72] dataset. The Swiss-EPFL dataset, which covers an expansive area of 8.18 square kilometers, derives its LoD2.0 models from data made publicly accessible by the Swiss federal authorities [9–11]. We illustrate the 3D LoD maps and query images of these two datasets in Figure 2. More details can be found in Appendix A, B and C.

**Baseline.** We compared our approach with two visual localization baselines: UAVD4L [72], predicated on textured mesh models, and CadLoc [44], predicated on LoD models, employing diverse feature extractors and matchers. Both baselines employ a keypoint-based strategy: 1) SIFT [41] descriptor with traditional Nearest Neighbor (NN) matching, 2) learning-based extractor SuperPoint (SPP) [25] with graph-based networks Superglue (SPG) [54], 3) detector-free matcher LoFTR [66] and 4) e-LoFTR [71], 5) dense feature matcher RoMA [26]. Additionally, considering the line structure of the LoD model, we apply three line-based algorithms for the CadLoc: 6) deep neural network SOLD$^2$ [45] for joint detection and description of line segments, 7) deep line segment detector DeepLSD [46] with line detector in SOLD$^2$, 8) DeepLSD with wireframe-based representation and dual-softmax matching method GlueStick [47]. Further details about the implementation of the baseline experiments can be found in Appendix E.

**Metrics.** We follow the standard localization evaluation procedure [68] and set recall thresholds of $(2m, 2°)$, $(3m, 3°)$, and $(5m, 5°)$.

Table 2: **Quantitative comparison results over the UAVD4L-LoD dataset.**

| Method | | in-Traj. | | | out-of-Traj. | | |
| --- | --- | --- | --- | --- | --- | --- | --- |
| | | 2m-2° | 3m-3° | 5m-5° | 2m-2° | 3m-3° | 5m-5° |
| Sensor Priors | | 0 | 0 | 4.3 | 0 | 0 | 0.36 |
| UAVD4L *Mesh model* | SIFT+NN | 73.13 | 78.62 | 80.42 | 82.39 | 85.13 | 86.36 |
| | SPP+SPG | 91.71 | 92.02 | 92.14 | 93.43 | 93.70 | 93.80 |
| | LoFTR | 84.98 | 88.09 | 88.90 | 91.56 | 92.02 | 92.11 |
| | e-LoFTR | 84.47 | 88.21 | 88.96 | 91.06 | 91.93 | 92.02 |
| | RoMA | **93.27** | **93.70** | 93.77 | 95.03 | 95.53 | 95.53 |
| CadLoc *LoD model* | SIFT+NN | 0 | 0 | 0 | 0 | 0 | 0 |
| | SPP+SPG | 0 | 0 | 0 | 0 | 0 | 0 |
| | LoFTR | 0 | 0 | 0 | 0 | 0 | 0 |
| | e-LoFTR | 0.37 | 0.87 | 1.31 | 0.41 | 0.78 | 1.37 |
| | RoMA | 2.18 | 2.87 | 3.68 | 6.93 | 8.76 | 10.40 |
| | SOLD2 | 0 | 0 | 0 | 0 | 0 | 0 |
| | DeepLSD+SOLD2 | 0 | 0 | 0 | 0 | 0 | 0 |
| | DeepLSD+GlueStick | 0 | 0 | 0 | 0 | 0 | 0 |
| **Ours** *LoD model* | no $NWE$ | 10.41 | 16.21 | 24.19 | 6.93 | 12.64 | 21.62 |
| | no $USR$ | 70.39 | 85.47 | 95.32 | 82.62 | 94.71 | 97.63 |
| | no $Refine$ | 51.31 | 76.06 | 86.78 | 74.27 | 97.95 | 99.36 |
| | **Full model** | 84.41 | 91.77 | **96.95** | **95.94** | **99.00** | **99.36** |

## 4.1 Implementation Details

During training, we set a random seed to limit 3D wireframe points $\{\mathbf{P}_i\}$ to $2,000$ points, and the pose sampling number $m_l(\mathrm{x}), m_l(\mathrm{y}), m_l(\mathrm{z}), m_l(\theta)$ for level $l = 1, 2, 3$ is uniformly set to $[13, 7, 3]$ due to constraints related to CUDA memory. The image size is $(512, 480)$ for the UAVD4L dataset and $(720, 480)$ for the Swiss-EPFL dataset. The pose sampling range at level 1 is set as $[10, 10, 30, 7.5]$ which refers to $[\mathrm{r}_p(x), \mathrm{r}_p(y), \mathrm{r}_p(z), \mathrm{r}_p(\theta)]$. For the UAVD4L-LoD dataset, we incorporate a subset of synthesized images from UAVD4L [72], which includes buildings, as training data. For Swiss-EPFL, we train the model by combining synthetic images *LHS* and real query images from the CrossLoc [74] project, following its data split pattern.

During inference, experiments are executed on real query images $\{\mathbf{I}_i\}$ derived from two datasets. We make the following changes, the discrete retrieval points from the 3D wireframe are sampled at an interval of 1 meter. The pose sampling number is increased to $[m_l(\mathrm{x}), m_l(\mathrm{y}), m_l(\mathrm{z}), m_l(\theta)] = [10, 10, 30, 8]$ for all levels. $\lambda$ is set as $0.8$. The training and inference of the entire network are executed using 2 NVIDIA RTX 4090 GPUs. Additionally, we employ four variations to validate the effectiveness of our method. Specifically, 1) -*no NWE* means no neural wireframe estimation, which extracts explicit line segments using DeepLSD [46] and constructs a distance field for each segment. It then replaces the cost function in Equations 2 and 7 with the distance function values, and solves for the pose using coarse-to-fine pose selection followed by Gauss-Newton refinement; 2) -*no USR* means a model without uncertainty sampling range estimation (Sec. 3.2); 3) -*no Refine* denotes a model without pose refinement (Sec. 3.3); and 4) full model is our proposed LoD-Loc.

## 4.2 Evaluation Results

**Evaluation over UAVD4L-LoD dataset.** As described in Table 2, our method shows excellent performance, both in the *in-Traj.* and *out-of-Traj.* queries. Apart from the $2m - 2°$ and $3m - 3°$ metric in the *in-Traj* queries, which are marginally lower than UAVD4L with RoMA matcher, all other metrics surpass those of contemporary baselines. Note that this comparison is unfair, as baselines reference on a high-precision texture model that is richer in texture and geometry, while we only employ a LoD model. We further compare with CadLoc, which shares the same 3D reference model as ours. However, we observe that regardless of the choice of descriptors (point-based or line-based), these methods perform poorly. We visualize their retrieval and matching failure cases in Appendix E.3. Furthermore, we analyze why our method performs better in the *out-of-Traj.* scenarios compared to the *in-Traj.* scenarios in Appendix F.3.

Table 3: **Quantitative comparison results over the Swiss-EPFL dataset**.

| Method | | *in-Place* | | | *out-of-Place* | | |
|---|---|---|---|---|---|---|---|
| | | 2m-2° | 3m-3° | 5m-5° | 2m-2° | 3m-3° | 5m-5° |
| Generated Priors | | 0 | 0 | 0.56 | 0 | 0 | 1.06 |
| UAVD4L *Mesh model* | SIFT+NN | 14.47 | 23.31 | 36.52 | 32.98 | 54.35 | 71.50 |
| | SPP+SPG | 34.83 | 60.39 | 77.25 | 77.04 | 89.71 | 92.35 |
| | LoFTR | 27.67 | 49.58 | 66.43 | 68.87 | 81.00 | 84.96 |
| | e-LoFTR | 37.64 | 60.96 | 76.40 | 81.53 | 91.03 | 93.93 |
| | RoMA | 45.98 | **66.77** | **80.73** | **89.18** | **98.68** | **98.94** |
| CadLoc *LoD model* | SIFT+NN | 0 | 0 | 0 | 0 | 0 | 0 |
| | SPP+SPG | 0 | 0 | 0 | 0 | 0 | 0 |
| | LoFTR | 0 | 0 | 0 | 0 | 0 | 0 |
| | e-LoFTR | 0 | 0.14 | 0.14 | 0 | 0 | 0.53 |
| | RoMA | 0.98 | 1.97 | 2.67 | 2.37 | 5.01 | 6.33 |
| | SOLD2 | 0 | 0 | 0 | 0 | 0 | 0 |
| | DeepLSD+SOLD2 | 0 | 0 | 0 | 0 | 0 | 0 |
| | DeepLSD+GlueStick | 0 | 0 | 0 | 0 | 0 | 0 |
| **Ours** *LoD model* | no $NWE$ | 11.37 | 21.35 | 33.57 | 18.99 | 31.39 | 45.91 |
| | no $USR$ | 42.42 | 58.29 | 71.21 | 31.40 | 48.81 | 70.45 |
| | no $Refine$ | 36.10 | 58.01 | 76.97 | 18.21 | 39.31 | 66.23 |
| | **Full model** | **48.60** | 65.31 | 79.78 | 37.73 | 57.26 | 77.57 |

Table 4: **Ablation study on different stages.** T.e./R.e. means translation/rotation error.

| Category | Level | Recall (%) | | | Median Error | |
|---|---|---|---|---|---|---|
| | | 2m-2° | 3m-3° | 5m-5° | T.e.(m) | R.e.(°) |
| *in-Traj.* | Level 1 | 23.88 | 60.35 | 83.85 | 2.58 | 1.41 |
| | Level 2 | 48.57 | 75.06 | 85.10 | 2.03 | 1.27 |
| | Level 3 | 51.31 | 76.06 | 86.78 | 1.97 | 1.25 |
| | Refine | **84.41** | **91.77** | **96.95** | **0.97** | **0.52** |
| *out-of-Traj.* | Level 1 | 34.81 | 78.01 | 97.67 | 2.31 | 1.05 |
| | Level 2 | 65.37 | 95.35 | 99.22 | 1.76 | 0.97 |
| | Level 3 | 74.27 | 97.95 | 99.36 | 1.63 | 0.95 |
| | Refine | **95.94** | **99.00** | **99.36** | **1.06** | **0.49** |

**Evaluation over Swiss-EPFL dataset.** Table 3 presents the inference results on the Swiss-EPFL dataset. CadLoc continues to exhibit widespread failures due to its poor retrieval and matching results across different modalities. We surpass the state-of-the-art UAVD4L method in the *in-Place* queries, but fall behind in the *out-of-Place* queries. Moreover, the overall results obtained on the Swiss-EPFL dataset are not as strong as those on the UAVD4L-LoD dataset. We attribute this discrepancy to the inferior quality of the images in the training database as explained in Appendix F.1. Besides, the LoD2.0 model provides less structured information, making it harder for pose inferences.

**Analysis of Methodological Advantages.** First, compared to the SOTA texture-based approach, the LoD-Loc employs distinct cues for localization. The texture-based method determines the pose by optimizing the re-projection error of corresponding 2D-3D points. Conversely, the LoD-Loc aligns the 3D wireframe projection to solve the pose. Second, 3D-model-based methods typically employ a two-stage scheme, which involves building 2D-3D matches and then solving the pose with PnP RANSAC. The LoD-Loc method directly solves the pose in an end-to-end manner, potentially leading to better pose accuracy. Third, the LoD-Loc includes several important modules to improve performance, such as coarse-to-fine pose cost volume reconstruction, uncertainty-based sampling range estimation, and differential Gauss-Newton refinement. These factors contribute to the superior performance of our method.

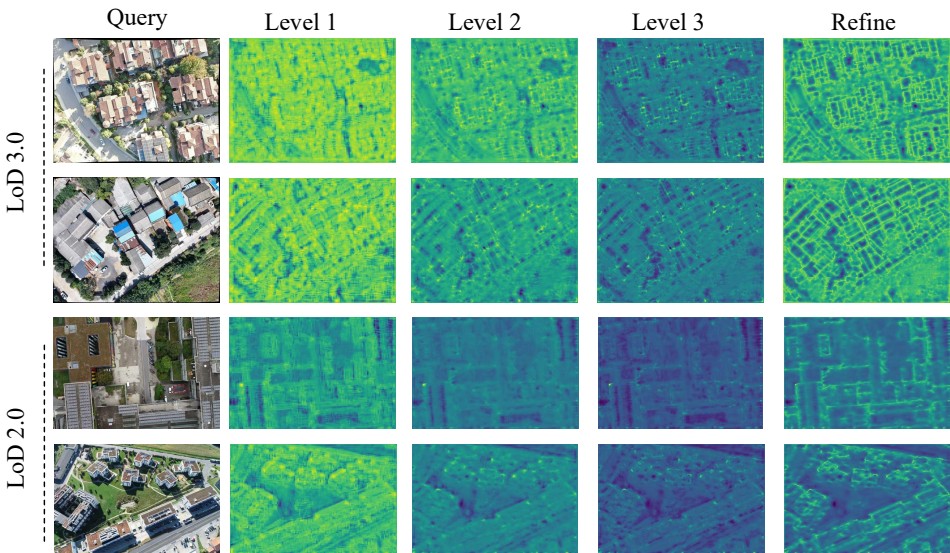

Figure 5: **Visualization of feature maps from different levels.** The feature maps of different levels reflect different fineness of wireframe extraction.

### 4.3 Ablation Studies

We perform ablation experiments on the UAVD4L-LoD dataset, focusing on different levels. More ablation studies are provided in Appendix F.2.

**Levels.** As depicted in Table 4, we present the results of the ablation experiments in terms of recall, translation errors, and rotation errors. The localization accuracy shows a gradual improvement as the number of levels increases. This demonstrates the effectiveness of the progressive coarse-to-fine estimation and final pose refinement. Figure 5 visualizes feature maps for each level, illustrating that wireframe features extracted from deeper levels is clearer.

## 5 Conclusion

This paper presents LoD-Loc, a novel approach for localizing aerial images using a LoD 3D map. Compared to large and expensive 3D maps that existing methods rely on, the LoD map provides a simple, accessible, and privacy-friendly scene representation. With the coarse sensor pose, the proposed LoD-Loc uses a unified pipeline to estimate the camera pose, including a multi-scale feature extractor, pose selection from cost volume, and pose refinement. Furthermore, we contribute two datasets with map levels of LoD3.0 and LoD2.0, along with real RGB queries with ground-truth pose annotation. LoD-Loc achieves excellent performance, even surpassing current state-of-the-art methods that use textured 3D models for localization. We believe LoD-Loc opens new possibilities for visual localization with simple and scalable 3D maps.

**Limitation.** LoD-Loc operates under the assumptions of a known gravity direction and a location prior. While these assumptions are reasonable, they restrict the application of LoD-Loc in environments where GPS is denied or unavailable.

**Broader impact.** This work has implications regarding privacy and surveillance. However, the LoD models represent building structures in a highly abstracted form, which alleviates concerns about the disclosure of personal privacy or land resource information.

## Acknowledgments and Disclosure of Funding

The authors would like to acknowledge the support from the National Natural Science Foundation of China (Grant No. 62406331).

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

# A  Details on Dataset Collection

The released datasets consist of two distinct parts, UAVD4L-LoD and Swiss-EPFL, providing LoD3.0 and LoD2.0 models, respectively. The UAVD4L-LoD dataset, which spans an area of 2.5 square kilometers, is derived from a semi-automatic process that generates a 3D LoD map based on the mesh model of the UAVD4L [72] dataset. This dataset includes a diverse array of architectural structures, including skyscrapers, villas, apartment complexes, educational institutions, and rural dwellings. The query images for this dataset were captured using two UAVs equipped with real sensor data: a DJI M300 [3] drone with an H20T [2] camera and a DJI Mavic3 Pro [5] drone. The Swiss-EPFL dataset, which covers an expansive area of 8.18 square kilometers, sources its LoD2.0 models from data made publicly accessible by the Swiss federal authorities [9–11]. This dataset features a variety of architectural styles, such as libraries, residential apartments, and stadiums. The query images for this dataset were acquired through the CrossLoc [74] projects, using a DJI Phantom 4 RTK [6] drone. Figure 2 presents the 3D LoD maps and query images from these two datasets.

## A.1  3D LoD Map Collection

The 3D LoD map for the UAVD4L-LoD dataset is generated semi-automatically with the assistance of the DP Modeler tool[4]. The process begins with the automatic generation of building blocks, characterized by their footprints and heights. Manual refinement is then applied to the architectural details of each building, raising them to the LoD3.0 level. The LoD accuracy of the UAVD4L-LoD dataset is consistent with the mesh model derived from UAVD4L.

For the Swiss-EPFL dataset, LoD2.0 models are downloaded from the SwissTOPO website [9–11]. To synchronize the coordinate systems between the map data and the drone-captured data from the CrossLoc dataset (which covers the same area with ground truth pose annotation), we converted the Swiss LoD map data in LV95 and LN02 coordinate systems to the ECEF coordinate system.

## A.2  Query Image Collection

The query images of the UAVD4L-LoD dataset are divided into two categories: *in-Traj.* and *out-of-Traj.*, representing trajectory-based and free-flight scenarios, respectively. The *in-Traj.* images, totaling $1,604$, were captured using a DJI M300 drone equipped with an H20T camera, focusing primarily on residential buildings, villas, and educational institutions. In contrast, the *out-Traj.* images, totaling $2,192$, were captured using a DJI Mavic3 Pro drone, covering a variety of architectural structures such as skyscrapers and rural dwellings. Both the *in-Traj.* and *out-of-Traj.* datasets include real sensor priors. Table 5 outlines the specific differences between the *in-Traj.* and *out-Traj.* sequences.

Table 5: **Key distinctions between the *in-Traj.* and *out-of-Traj.* sequences**.

| Name | Capture device | Capture pitch angle | Capture height | Capture route |
|------|----------------|---------------------|----------------|---------------|
| *in-Traj.* | DJI M300+H20t | $0°$ or $45°$ | 120m | Zig-zag flight on a selected region |
| *out-of-Traj.* | DJI Mavic3 Pro | $30° \sim 60°$ | $90m \sim 150m$ | Manually controlled flight on the map |

The real query images in the Swiss-EPFL dataset come from the CrossLoc [74] dataset. However, because the real-time kinematics (RTK) data from the DJI Phantom4 were used directly as ground truth (GT) poses, some GT poses show significant mislabeling. To resolve this issue, we projected the wireframes of LoD maps onto query images to identify and remove incorrectly labeled poses. The final query dataset comprises $2,254$ images.

## A.3  Query GT Generation

For the UAVD4L-LoD dataset, we employ a semi-automatic annotation approach to generate pseudo-GT poses $\{\bar{\xi}_i\}$ for the query images $\{\mathbf{I}_i^q\}$. The process is based on the SfM results and textured mesh model of the UAVD4L [72]. First, we perform SfM separately on the query images $\{\mathbf{I}_i^q\}$ and the reference images $\{\mathbf{I}_i^r\}$ from the UAVD4L to obtain SfM results $\mathcal{C}_q$ and $\mathcal{C}_r$. Next, we manually

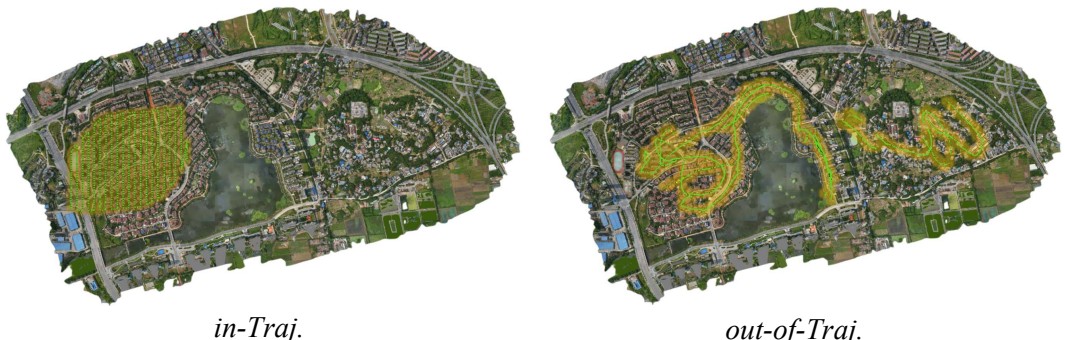



*in-Traj.*            *out-of-Traj.*



Figure 6: **Flight trajectories of query images in the UAVD4L-LoD dataset**. We present the flight trajectories of the registered $in\text{-}Traj.$ and $out\text{-}of\text{-}Traj.$ query images. The $in\text{-}Traj.$ images follow a predetermined flight path, primarily covering the left half of the map. In contrast, the $out\text{-}of\text{-}Traj.$ images navigate arbitrarily without a fixed route, randomly covering the entire map.

select points with distinctive visual features (e.g., building corners) as tie points to align $\mathcal{C}_q$ with $\mathcal{C}_r$. To further enhance the accuracy of the pseudo-GT, we utilize the render-and-compare pipeline [76] to refine the final poses $\{\bar{\xi}_i\}$.

Additionally, we analyze the discrepancies between the pose prior and the GT pose, decoupling the poses into 3D translation in WGS84 space and Euler angles in terms of 'yaw-pitch-roll'. It is observed that the translation errors for x and y are within $\pm10$, z errors are within $\pm30$, yaw errors are within $\pm7.5$, and pitch and roll errors are approximately 1 degree.

## B    Details on UAVD4L-LoD Dataset

### B.1    Pseudo-GT Generation

In the UAVD4L-LoD dataset, we employed a semi-automatic annotation technique to generate pseudo-GT poses $\{\bar{\xi}_i\}$ for the query images $\{\mathbf{I}_i\}$. Initially, we performed SfM separately on the query images $\{\mathbf{I}_i\}$ and the reference images $\{\mathbf{I}_j^r\}$ from UAVD4L, yielding corresponding SfM results $\mathcal{C}_q$ and $\mathcal{C}_r$. Subsequently, based on the capture region of the $\{\mathbf{I}_i\}$, we manually identified $e$ distinctive tie points, such as the corner of the building, to align $\mathcal{C}_q$ with $\mathcal{C}_r$, resulting in $\mathcal{C}_f$. We then refined the pose accuracy of $\mathcal{C}_f$ using Bundle Adjustment. The accuracy of the GT poses was evaluated through the median reprojection error, which was $0.43$ pixels for all connected points and $1.19$ pixels for the tie points. Finally, we employed a render-and-compare [76] pipeline for the final refinement of the GT poses. In this manner, with the annotation of tens of $e = 20$ manual tie points, we were able to obtain pseudo-GT poses $\{\bar{\xi}_i\}$ for a total of 3, 796 query images $\{\mathbf{I}_i\}$. Figure 6 shows the flight trajectories of the $in\text{-}Traj.$ and $out\text{-}of\text{-}Traj.$

### B.2    Sensor Pose Accuracy

In the UAVD4L-LoD dataset, we conduct a comprehensive data analysis to validate the precision of the sensor pose. This is accomplished by employing absolute error bar charts, as illustrated in Figure 7. Additionally, we assess the accuracy by projecting wireframe points onto the image plane using both sensor and GT poses. Results of these projections are depicted in Figure 8.

## C    Details on Swiss-EPFL Dataset

### C.1    Data Cleaning

In the Swiss-EPFL dataset, the GT poses $\{\bar{\xi}_i\}$ for the query images $\{\mathbf{I}_i\}$ are sourced from the CrossLoc project [74]. This project directly acquires RTK data from the DJI Phantom 4 for GT annotation. Considering that the RTK device may introduce some noise, we identified and excluded query images with incorrect labeling. This was accomplished by projecting the wireframe onto the

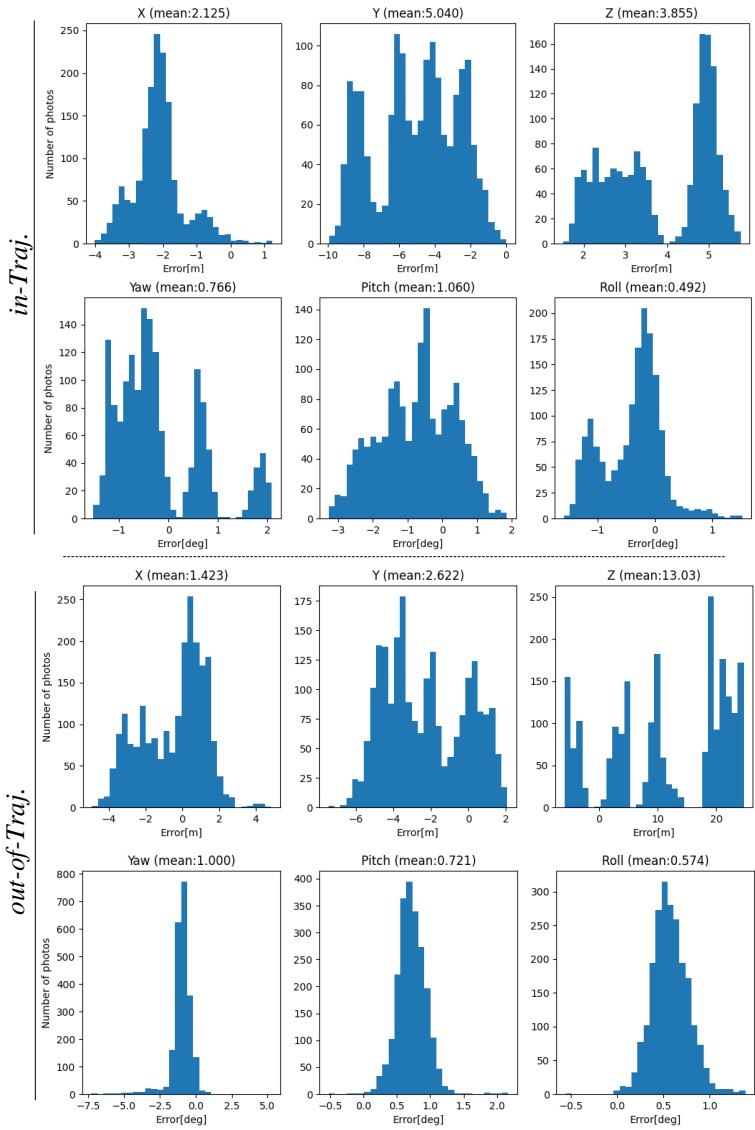

Figure 7: **The errors between priors and GT poses**. We visualize the absolute pose errors between the sensor and GT pose in 6-DoF. The errors in the $X$-$Y$-$Z$-$yaw$ dimensions show indicate substantial discrepancies. Specifically, the errors in the $X$-$Y$ range from $-10$ to $10$ meters, the $Z$ ranges from $-30$ to $30$ meters, and the $yaw$ fluctuates within the range of $-7.5$ to $7.5$ degrees.

image plane and manually discarding the items exhibiting noticeable misalignment. The process is visualized in Figure 9.

## C.2 Sensor Poses Generation

Since the CrossLoc [74] project does not provide GPS or other sensor data, we randomly generate sensor poses $\boldsymbol{\xi}_p$ by emulating the pose errors derived from the UAVD4L-LoD dataset. Specifically, $X$-$Y$ for position range between $[-10, 10]$ meters, $Z$ ranges between $[-30, 30]$ meters, $yaw$ for rotation ranges in $[-7.5, 7.5]$ degrees, and $pitch$-$roll$ range between $[-1, 1]$ degrees. We present the discrepancy between the generated sensor poses and GT poses in a bar chart, as depicted in Figure 11.

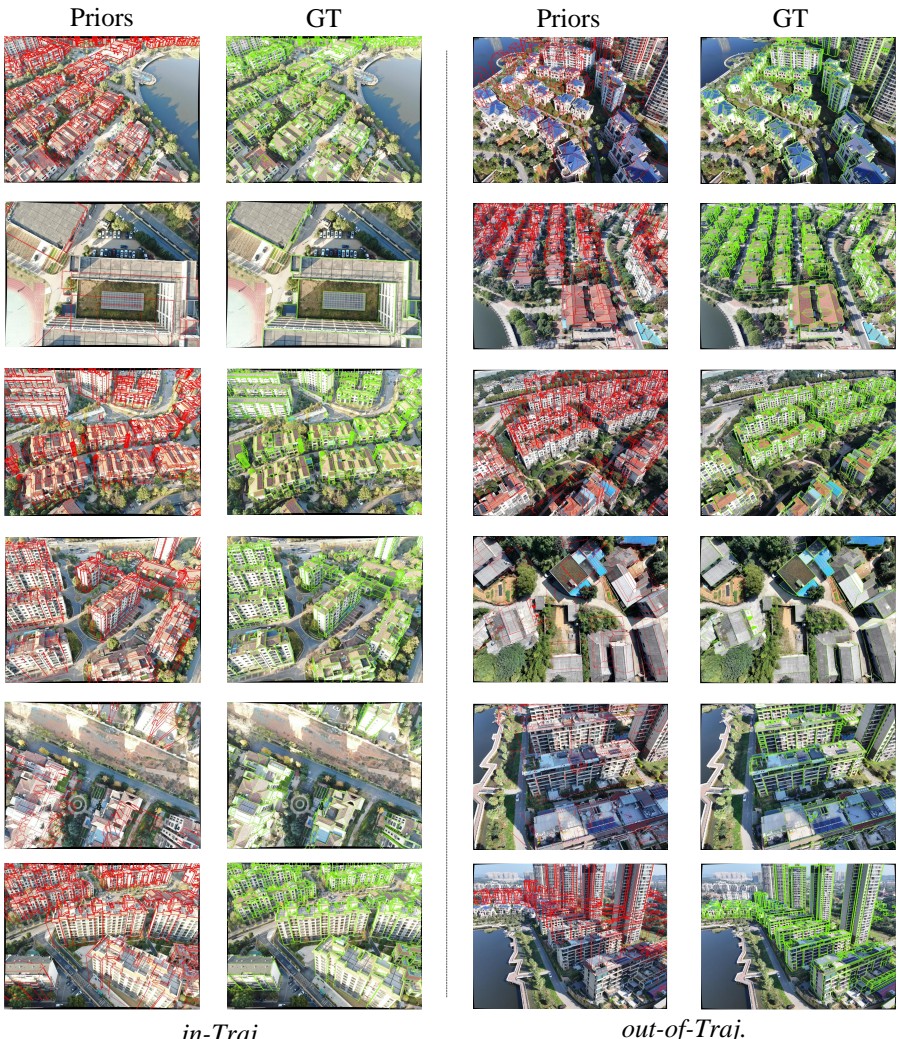

| Priors | GT | Priors | GT |

*in-Traj.*           *out-of-Traj.*

Figure 8: **3D wireframe projection over UAVD4L-Lod dataset.** We visualize the projected wireframe on query images based on sensor and GT poses to demonstrate their accuracy.

# D  Details on Method

## D.1  Architecture of Multi-scale Feature Extractor

In this section, we provide a detailed description of the architecture of the multi-scale feature extractor in Table 6.

## D.2  Jacobian Computation

The objective function for pose refinement is:

$$E(\boldsymbol{\xi}^*) = -\sum_i ||f_i||^2 = -\sum_i ||\mathbf{F}_{rf}[\Pi(\mathbf{R}^* \cdot \mathbf{P}_i + \mathbf{t}^*)]||^2. \tag{11}$$

We compute the Jacobian matrix of the residual function $f_i$ with respect to the pose parameters as followed:

$$J_i = \frac{\partial f_i}{\partial \xi^*} = \frac{\partial \mathbf{F}_{rf}}{\partial p_i} \frac{\partial p_i}{\partial \mathbf{P}_i^{cam}} \frac{\partial \mathbf{P}_i^{cam}}{\partial \xi^*}, \tag{12}$$

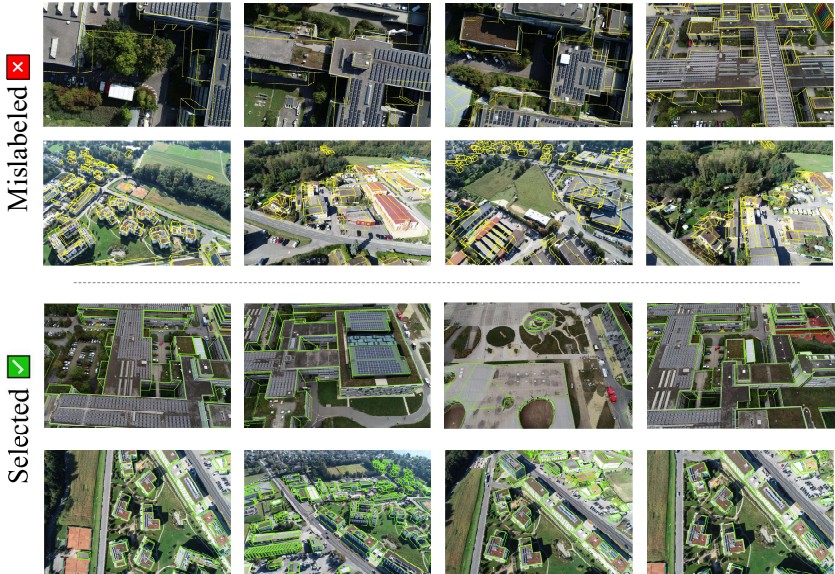

Figure 9: **Samples of mislabeled and selected query images over Swiss-EPFL dataset**. We eliminate mislabeled query images by manually identifying the alignment between the projected 2D wireframe and the corresponding RGB image.

where $\frac{\partial \mathbf{F}_{rf}}{\partial p_i}$ is the gradient of the feature map $\mathbf{F}_{rf}$ at the 2D location $p_i$ and

$$
\begin{aligned}
p_i &= \Pi(\mathbf{P}_i^{cam}) = \begin{pmatrix} \frac{X_i^{cam}}{Z_i^{cam}} f_x + c_x \\ \frac{Y_i^{cam}}{Z_i^{cam}} f_y + c_y \end{pmatrix}, \\
\frac{\partial p_i}{\partial \mathbf{P}_i^{cam}} &= \begin{pmatrix} \frac{1}{Z_i^{cam}} f_x & 0 & -\frac{X_i^{cam}}{(Z_i^{cam})^2} f_x \\ 0 & \frac{1}{Z_i^{cam}} f_y & -\frac{Y_i^{cam}}{(Z_i^{cam})^2} f_y \end{pmatrix}.
\end{aligned}
\tag{13}
$$

Besides, $\mathbf{P}_i^{cam}$ is the point which transformed to the camera space. To compute the last derivative of Eq. 12, we add a perturbation $\Delta \xi$ to the transformation:

$$
\mathbf{P}_i^{cam} = \mathbf{R}^*(\Delta \mathbf{R} \mathbf{P}_i + \Delta \mathbf{t}) + \mathbf{t}^*,
\tag{14}
$$

Finally, the derivatives w.r.t the translation component and rotation component are:

$$
\begin{aligned}
\frac{\partial \mathbf{P}_i}{\partial \xi_t^*} &= \frac{\partial \mathbf{P}_i}{\partial \Delta \mathbf{t}} = \mathbf{R}^* \\
\frac{\partial \mathbf{P}_i}{\partial \xi_r^*} &= \frac{\partial \mathbf{P}_i}{\partial \Delta \mathbf{R}} = -\mathbf{R}^*[\mathbf{P}_i]_\times,
\end{aligned}
\tag{15}
$$

where $[]_\times$ is the skew-symmetric matrix.

# E  Details on Baseline

## E.1  Sensor-guided Image Retrieval

For baselines, a retrieval-and-matching process is used upon the reference images in the dataset. To ensure a fair comparison, we apply the sensor poses to guide the image retrieval process for UAVD4L [72] and Cad-Loc [44]. For each query image $\mathbf{I}$, we narrow the retrieval candidates $^q\mathcal{I}$ using Eq. 16.

$$
^q\mathcal{I} = \{ \mathbf{I}_i^r \mid \forall \ \| \mathbf{t}_i^r - \mathbf{t}^q \| \leq \gamma_t, \arccos(\mathbf{R}_i^r, \mathbf{R}^q) \leq \gamma_o \},
\tag{16}
$$

where $\|\cdot\|$ denotes the Frobenius Norm between two translation matrices, $\arccos(\cdot)$ calculates the rotation angles between two matrices, $\gamma_t$ and $\gamma_o$ are the threshold for translation and orientation, respectively. To determine the proper values for $\gamma_t$ and $\gamma_o$ for the baseline methods, a series of

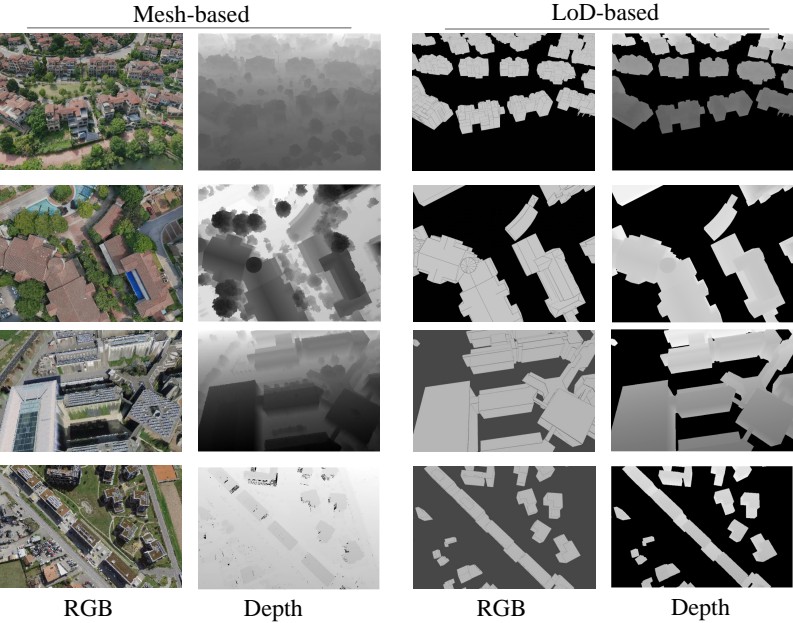

Figure 10: **Visualization of reference RGB and depth maps.** RGB and depth maps are rendered using a textured mesh model or a 3D LoD map.

experiments are conducted on the UAVD4L dataset. In these experiments, we hypothesize that if no reference image could be located within the defined search area, the sensor pose would be utilized as the localization result. Table 7 shows that stricter thresholds result in worse outcomes. Consequently, we set $\gamma_t = 150$ and $\gamma_o = 30$ to ensure a sufficient search space. Furthermore, we measure the impact of retrieval number $k$ in Table 8. The results suggest that while a larger $k$ value enhances the performance of the benchmark algorithm, it also leads to an increase in inference time. Following previous work [72], the retrieval number is set at $k = 3$. It is worth noting that regardless of the choice of $k$, our method exhibits a substantial acceleration in speed, outperforming by several-fold, or even an order of magnitude.

## E.2 Reference Image Details

We provide a detailed description of the reference images used in the two datasets. These images serve dual purposes: they function as the database images for retrieval and matching in baselines, and they are also utilized as training data for the proposed LoD-Loc method. Specifically, for the UAVD4L-LoD dataset, we use a subset dataset of synthesis images in UAVD4L [72], excluding data that does not contain buildings. For the Swiss-EPFL dataset, synthetic images rendered in Latin Hypercube Sampling (LHS) [74] pattern have been employed as reference images. Notably, the CrossLoc dataset [74] did not include images in proximity to the *out-of-Place* area. To address this, we adopted the same synthetic scheme from [74] to generate synthetic reference images for this region. Figure 10 shows reference samples of $RGB$ images and $Depth$ images for both the mesh-based model and LoD-based model.

## E.3 Failure Cases in Baselines

Although baselines have achieved impressive performance, they suffer from retrieving and matching repetitive texture images and cross-modal images. For example, Figure 13 exhibits deficiencies in retrieving repetitive texture images, and Figure 14 depicts poor matching results between RGB and LoD-rendered images.

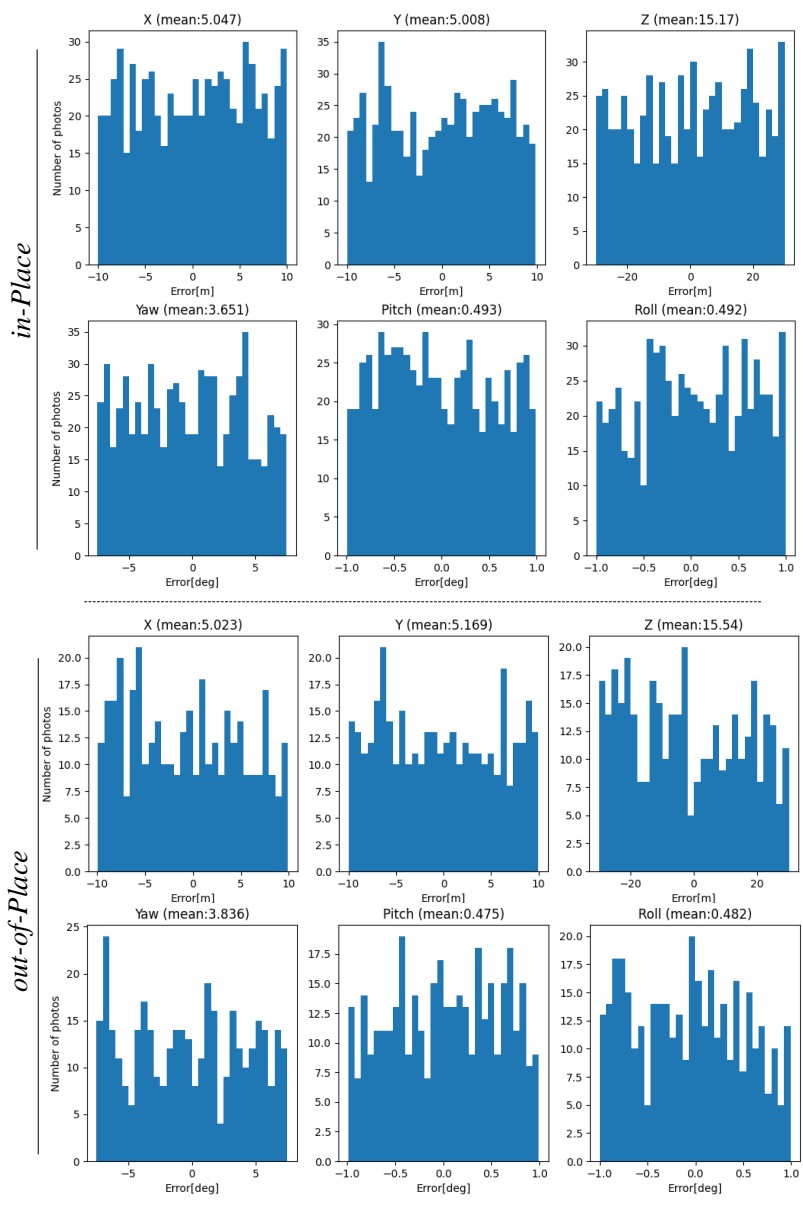

Figure 11: **The discrepancy between our generated poses and GT poses over the Swiss-EPFL dataset.** We use the generated poses to simulate the pose of the sensor.

# F    Details of Experiments

## F.1    Visualization of Training Data

We visualize some synthetic training samples of LoD-Loc, as shown in Figure 15. For the Swiss-EPFL dataset, the reference 3D model is derived from LiDAR point clouds, Terrain Models, and Orthophotos. In contrast, in the UAVD4L-LoD dataset, the reference 3D model is generated from

Table 6: **The architecture of our multi-scale feature extractor**. We discuss the details of each convolutional unit. $conv$ represents a unit consisting of a 2D convolutional layer, a batch normalization layer and a ReLU layer. While $fine\_conv$ denotes a general convolutional layer. $deconv$ means a deconvolutional unit. The colored cells are the outputs for each level $l$ with a single channel.

| Layer | Stride | Kernel | Channel | Input |
|---|---|---|---|---|
| conv0_0 | 1×1 | 3×3 | 3→8 | rgb |
| conv0_1 | 1×1 | 3×3 | 8→8 | conv0_0 |
| conv1_0 | 2×2 | 5×5 | 8→16 | conv0_1 |
| conv1_1 | 1×1 | 3×3 | 16→16 | conv1_0 |
| conv1_2 | 1×1 | 3×3 | 16→16 | conv1_1 |
| conv2_0 | 2×2 | 5×5 | 16→32 | conv1_2 |
| conv2_1 | 1×1 | 3×3 | 32→32 | conv2_0 |
| conv2_2 | 1×1 | 3×3 | 32→32 | conv2_1 |
| conv_out1 | 1×1 | 1×1 | 32→1 | conv2_2 |
| deconv1_0 | 2×2 | 3×2 | 32→16 | conv2_2 |
| concat1 | - | - | - | deconv1_0, conv1_2 |
| conv3_0 | 1×1 | 3×3 | 32→16 | concat1 |
| conv_out2 | 1×1 | 1×1 | 16→1 | conv3_0 |
| deconv2_0 | 2×2 | 3×3 | 16→8 | conv3_0 |
| concat2 | - | - | - | deconv2_0, conv0_1 |
| conv4_0 | 1×1 | 3×3 | 16→8 | concat2 |
| conv_out3 | 1×1 | 1×1 | 8→1 | conv4_0 |
| concat3 | - | - | - | conv4_0, conv_out3, rgb |
| fine_conv0 | 1×1 | 5×5 | 12→24 | concat3 |
| fine_conv1 | 1×1 | 5×5 | 24→12 | fine_conv0 |
| conv_out4 | 1×1 | 1×1 | 12→1 | fine_conv1 |

Table 7: **Ablation study on different threshold $\gamma_t$ and $\gamma_o$ for baselines**.

| Method | | Threshold $(\gamma_t, \gamma_o)$ | in-Traj. | | | out-of-Traj. | | |
|---|---|---|---|---|---|---|---|---|
| | | | 2m-2° | 3m-3° | 5m-5° | 2m-2° | 3m-3° | 5m-5° |
| UAVD4L | SIFT+NN | (30, 7.5) | 0.62 | 0.69 | 4.99 | 25.87 | 26.82 | 27.42 |
| | | (50, 15) | 27.00 | 28.30 | 32.29 | 55.66 | 57.44 | 58.26 |
| | | (150, 30) | **73.13** | **78.62** | **80.42** | **82.39** | **85.13** | **86.36** |
| | SPP+SPG | (30, 7.5) | 0.94 | 0.94 | 5.24 | 30.11 | 30.20 | 30.29 |
| | | (50, 15) | 33.92 | 33.92 | 37.28 | 60.99 | 61.13 | 61.18 |
| | | (150, 30) | **91.71** | **92.02** | **92.14** | **93.43** | **93.70** | **93.80** |
| | LoFTR | (30, 7.5) | 0.94 | 0.94 | 5.20 | 29.79 | 30.02 | 30.16 |
| | | (50, 15) | 33.29 | 33.35 | 36.72 | 60.90 | 60.90 | 60.99 |
| | | (150, 30) | **84.98** | **88.09** | **88.90** | **91.56** | **92.02** | **92.11** |

high-resolution aerial imagery through oblique photography reconstruction. As a result, the synthetic images from the former are of a lower quality. This could partly elucidate why our method yields lower results on the Swiss-EPFL dataset compared to UAVD4L-LoD.

## F.2 Additional Ablation Studies

We provide more ablation studies in this section, which include the pose sampling number, the sample density $\delta$ of 3D wireframes, the sampling range controller lambda $\lambda$. Additionally, we explore the convergence and generalization of our method.

**Pose sampling number.** As illustrated in Table 9, we report the experimental results with varying numbers of pose samples. The findings suggest that a reduction in the number of sampled poses brings about a decrease in accuracy.

Table 8: **Ablation study on different Top-$k$ for baselines**.

| Method | | Top-$k$ | *in-Traj.* | | | *out-of-Traj.* | | | Time (s) |
|---|---|---|---|---|---|---|---|---|---|
| | | | 2m-2° | 3m-3° | 5m-5° | 2m-2° | 3m-3° | 5m-5° | |
| UAVD4L | SIFT+NN | 3 | 73.13 | 78.62 | 80.42 | 82.39 | 85.13 | 86.36 | 1.85 |
| | | 10 | 85.97 | 89.65 | 90.52 | 90.28 | 92.43 | 93.66 | 1.96 |
| | | 20 | 88.09 | 91.33 | 92.64 | 92.75 | 94.71 | 95.99 | 2.13 |
| | SPP+SPG | 3 | 91.71 | 92.02 | 92.14 | 93.43 | 93.70 | 93.80 | 1.79 |
| | | 10 | 99.25 | 99.31 | 99.31 | 98.45 | 98.49 | 98.49 | 3.31 |
| | | 20 | 99.75 | 99.81 | 99.81 | 99.91 | 99.95 | 99.95 | 5.44 |
| | LoFTR | 3 | 84.98 | 88.09 | 88.90 | 91.56 | 92.02 | 92.11 | 1.70 |
| | | 10 | 90.21 | 91.65 | 92.08 | 94.75 | 94.89 | 94.89 | 3.78 |
| | | 20 | 85.97 | 87.53 | 87.91 | 90.37 | 90.83 | 91.29 | 6.26 |
| **Ours** | | − | 84.41 | 91.77 | 96.95 | 95.94 | 99.00 | 99.36 | 0.34 |

Table 9: **Ablation study on different pose sampling numbers for LoD-Loc.**

| Category | Numbers on $[\theta, x, y, z]$ | Recall (%) | | | Median Error | |
|---|---|---|---|---|---|---|
| | | 2m-2° | 3m-3° | 5m-5° | T.e. (m) | R.e. (°) |
| *in-Traj.* | [2 , 3 , 3 , 8] | 18.83 | 24.94 | 36.03 | 7.67 | 4.37 |
| | [4, 5 , 5 , 15] | 77.68 | 84.98 | 90.15 | 1.07 | 0.59 |
| | [8, 10, 10, 30] | **84.41** | **91.77** | **96.95** | **0.97** | **0.52** |
| *out-of-Traj.* | [2 , 3 , 3 , 8] | 12.36 | 16.93 | 23.81 | 11.49 | 5.51 |
| | [4, 5 , 5 , 15] | 87.27 | 93.25 | 94.25 | 1.15 | 0.54 |
| | [8, 10, 10, 30] | **95.94** | **99.00** | **99.36** | **1.06** | **0.49** |

**3D wireframe points sampling density.** We conduct ablation studies for varying sampling densities, which affects the interpolation process on the feature map. As depicted in Table 10, there is no significant fluctuation in localization accuracy with changes in sampling density.

**Sampling range controller.** The parameter lambda $\lambda$ adjusts the length of the sampling range. Through ablation studies, we demonstrate that the sensitivity of this parameter during the testing phase is low. The results are shown in Table 11.

**Convergence and initial poses.** Table 12 reports the localization recall with different initial prior errors on the UAVD4L-LoD dataset. It can be observed that the success rate of localization decreases as the initial prior error increases. Such issues occur when the GPS signal in the air is heavily interfered with. In such cases, we believe using sequence information could be a possible solution.

**Cross-scene generalization.** Table 13 illustrates the generalization capability of LoD-Loc through training and testing in diverse regions. Figure 16 delineates regional data using distinctive symbols and colors. On the UAVD4L-LoD dataset (A1 and A2), cross-scene testing yields results slightly lower than those obtained from training on the entire scene. For the Swiss-EPFL dataset (B1 and B2), we employ a model trained on the synthetic UAVD4L-LoD dataset, which achieves similar or even better performance compared to a model trained specifically on the Swiss-EPFL dataset. Additionally, the supplementary materials include two demo videos showcasing the model's capacity to localize cross-modal thermal images.

**Computational cost comparison.** We conducted test experiments on a single batch (Batch Size = 1) of images using the NVIDIA GeForce RTX 4090 device, and recorded the average peak CUDA usage as well as the average inference time. The details are provided in Table 14

### F.3 Visualization of Results

We present more visualization results, including examples of corner houses (Figure 12), feature maps (Figure 17) and prediction results (Figure 18) at different levels. We found that the preset zig-zag route in a selected region resulted in some images capturing only the corners of houses, as shown in Figure 18. This led to poorer performance under strict 2m-2° metrics. However, it is important

Table 10: **Ablation study on different wireframe sampling density**. x-$m$ means sampling per-x meter on each wireframes.

| | Category | Density $\delta$ | Recall (%) | | | Median Error | |
|---|---|---|---|---|---|---|---|
| | | | 2m-2° | 3m-3° | 5m-5° | T.e. (m) | R.e. (°) |
| LoD-Loc | *in-Traj.* | 4-$m$ | 85.10 | 92.39 | 96.51 | 0.95 | 0.52 |
| | | 2-$m$ | 84.16 | 91.08 | 96.95 | 0.97 | 0.52 |
| | | 1-$m$ | 84.41 | 91.77 | 96.95 | 0.97 | 0.52 |
| | *out-of-Traj.* | 4-$m$ | 95.21 | 98.68 | 99.18 | 1.00 | 0.45 |
| | | 2-$m$ | 95.44 | 98.91 | 99.32 | 1.06 | 0.48 |
| | | 1-$m$ | 95.94 | 99.00 | 99.36 | 1.06 | 0.49 |

Table 11: **Ablation study on different Lambda $\lambda$.**

| | Category | Lambda $\lambda$ | Recall (%) | | | Median Error | |
|---|---|---|---|---|---|---|---|
| | | | 2m2° | 3m3° | 5m5° | T.e. (m) | R.e. (°) |
| LoD-Loc | *in-Traj.* | 1.5 | 83.42 | 91.02 | 96.57 | 1.00 | 0.49 |
| | | 1 | 84.41 | 91.77 | 97.01 | 0.95 | 0.53 |
| | | 0.8 | 84.41 | 91.77 | 96.95 | 0.97 | 0.52 |
| | | 0.5 | 84.04 | 91.58 | 96.45 | 0.97 | 0.52 |
| | *out-of-Traj.* | 1.5 | 91.97 | 97.54 | 98.45 | 1.11 | 0.53 |
| | | 1 | 95.71 | 99.04 | 99.36 | 1.07 | 0.50 |
| | | 0.8 | 95.94 | 99.00 | 99.36 | 1.06 | 0.49 |
| | | 0.5 | 95.71 | 98.86 | 99.32 | 1.06 | 0.49 |

to note that in the *in-Traj.* scenario, our method achieves comparable or superior results for coarse metrics. For instance, we achieve 96.95% on 5m-5° while the closest baseline achieves 92.14%.

Table 12: **Impact of the initial pose for LoD-Loc.** The parameters $\Delta x$ and $\Delta y$ denote the error range in the horizontal plane, while $\Delta z$ represents the error range in the vertical dimension. For instance, $\Delta x = 10$ implies that the initial error in the $x$ value lies within the interval [-10, 10]. The rotation error remains consistent with the real sensor data. All error ranges are measured in meters.

| | Category | Prior Error Range $[\Delta x, \Delta y, \Delta z]$ | Recall (%) | | |
| --- | --- | --- | --- | --- | --- |
| | | | 2m-2° | 3m-3° | 5m-5° |
| LoD-Loc | *in-Traj.* | [10, 10, 30] | 84.41 | 91.77 | 96.95 |
| | | [20, 20, 30] | 87.28 | 90.77 | 91.65 |
| | | [30, 30, 30] | 78.93 | 82.98 | 83.85 |
| | | [50, 50, 30] | 43.08 | 48.82 | 50.69 |
| | | [100, 100, 30] | 5.67 | 7.36 | 8.79 |
| | *out-of-Traj.* | [10, 10, 30] | 95.94 | 99.00 | 99.36 |
| | | [20, 20, 30] | 82.07 | 88.05 | 89.55 |
| | | [30, 30, 30] | 74.27 | 80.66 | 81.79 |
| | | [50, 50, 30] | 46.53 | 53.60 | 55.98 |
| | | [100, 100, 30] | 6.93 | 9.95 | 11.99 |

Table 13: **Cross-scene generalization**. We assess the generalization ability of our method by training and testing on different regions. The regional divisions are illustrated in Figure 16, identified by a specific color and letter.

| | Train region *Synthesis* | Test region *Real* | Recall (%) | | |
| --- | --- | --- | --- | --- | --- |
| | | | 2m-2° | 3m-3° | 5m-5° |
| LoD-Loc | A2 | A1 | 83.39 | 91.50 | 96.81 |
| | A1, A2 | A1 | **89.51** | **95.01** | **97.98** |
| | A1 | A2 | 82.54 | 91.01 | 91.52 |
| | A1, A2 | A2 | **95.56** | **98.66** | **99.38** |
| | A1, A2 | B1 | **55.41** | **71.77** | **84.17** |
| | B1, B2 | B1 | 37.73 | 57.26 | 77.57 |
| | A1, A2 | B2 | **50.00** | 59.27 | 65.45 |
| | B1, B2 | B2 | 48.60 | **65.31** | **79.78** |

| | Method | Memory (Mb) | Time (s) |
| --- | --- | --- | --- |
| UAVD4L | SPP | 610 | 1.79 |
| | SIFT | 443 | 1.85 |
| | LoFTR | 2631 | 1.70 |
| | RoMA | 5488 | 4.68 |
| | eLoFTR | 1650 | 1.06 |
| ours | | 4810 | 0.34 |

Table 14: **Computational cost comparison.**

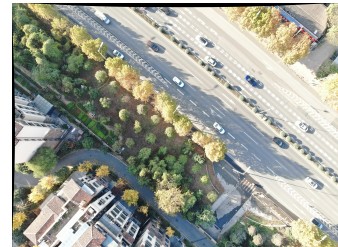

Figure 12: **Example of corner houses.**

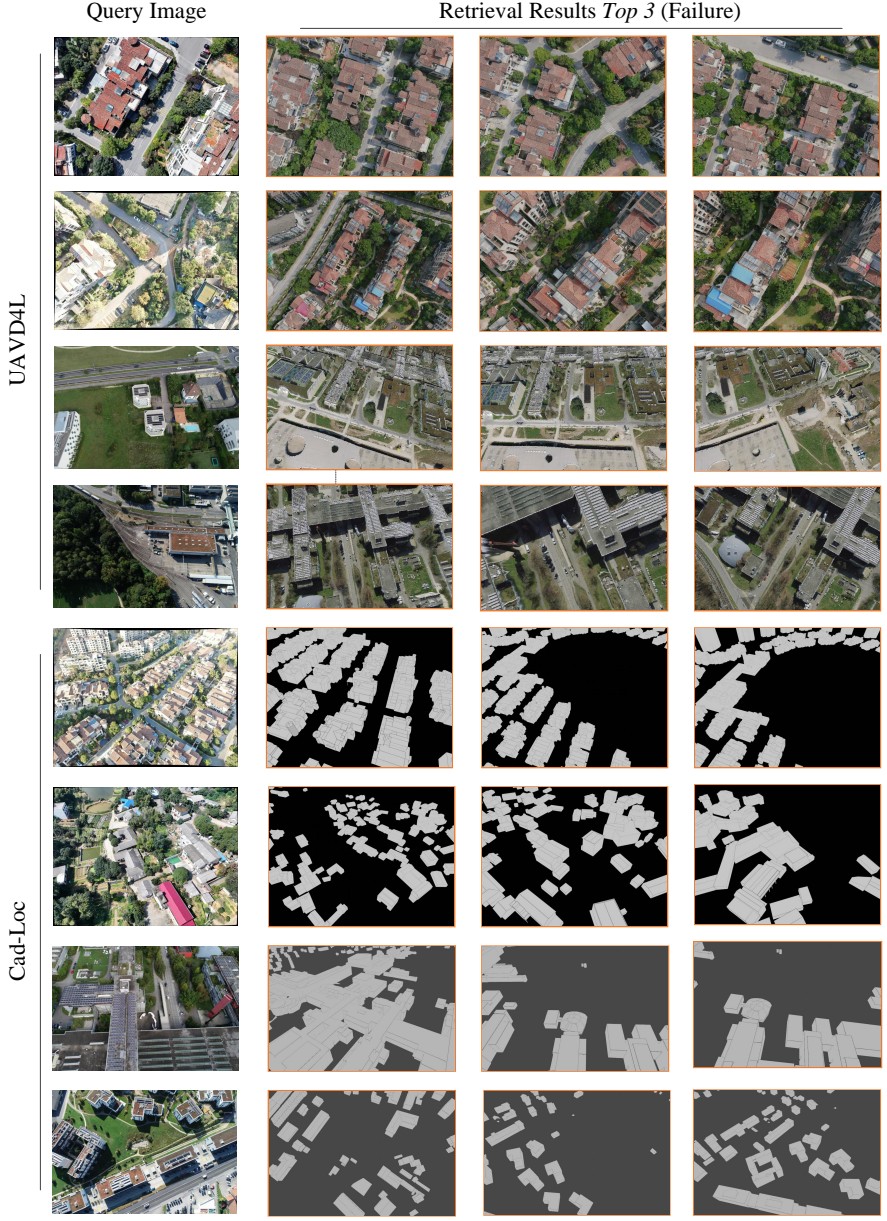

Figure 13: **Failure retrieval cases of baselines**. Even with narrowed searching scopes, the retrieval phase still suffers from issues such as repetitive textures and cross-modal challenges.

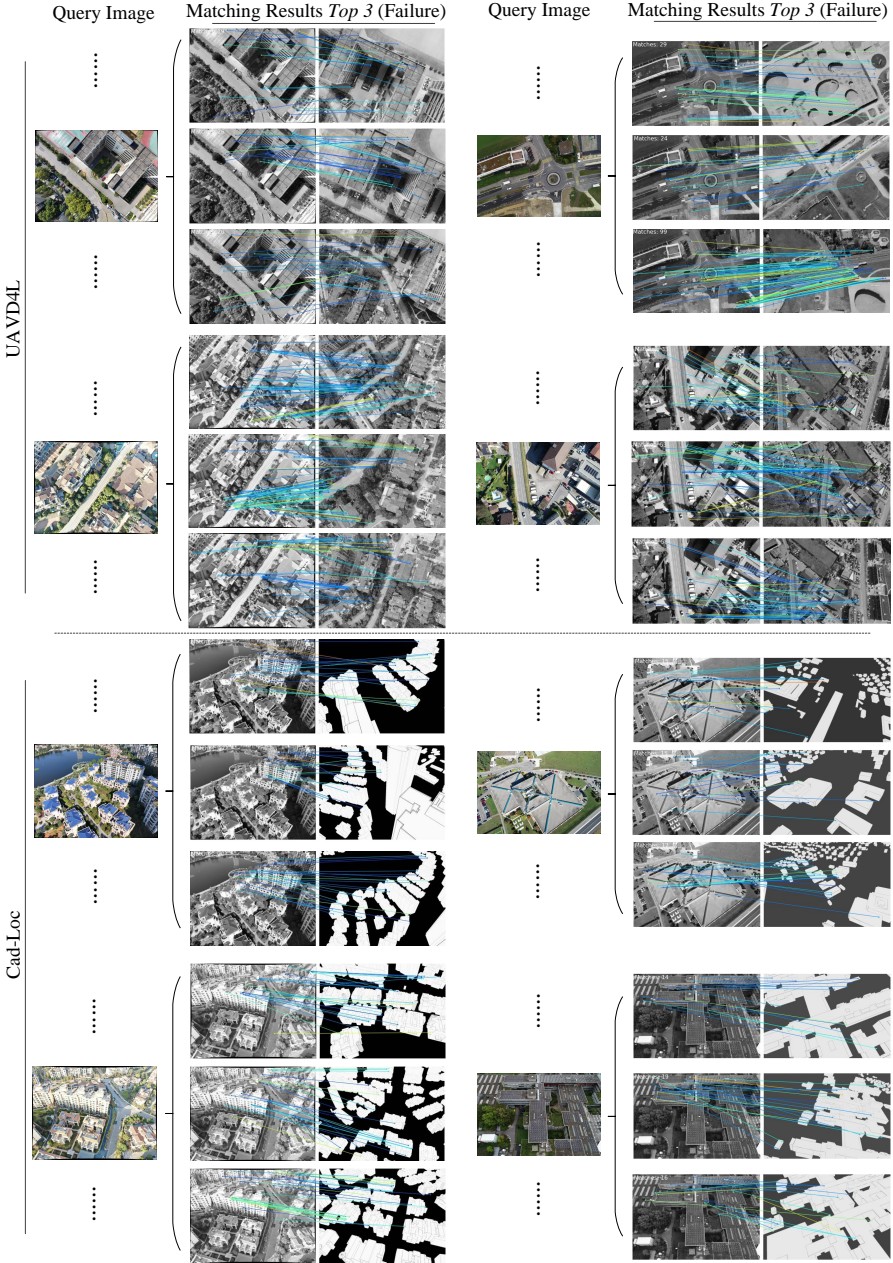

Figure 14: **Failure matching cases of baselines**. The differences in viewpoint and modality influence the results for image matching.

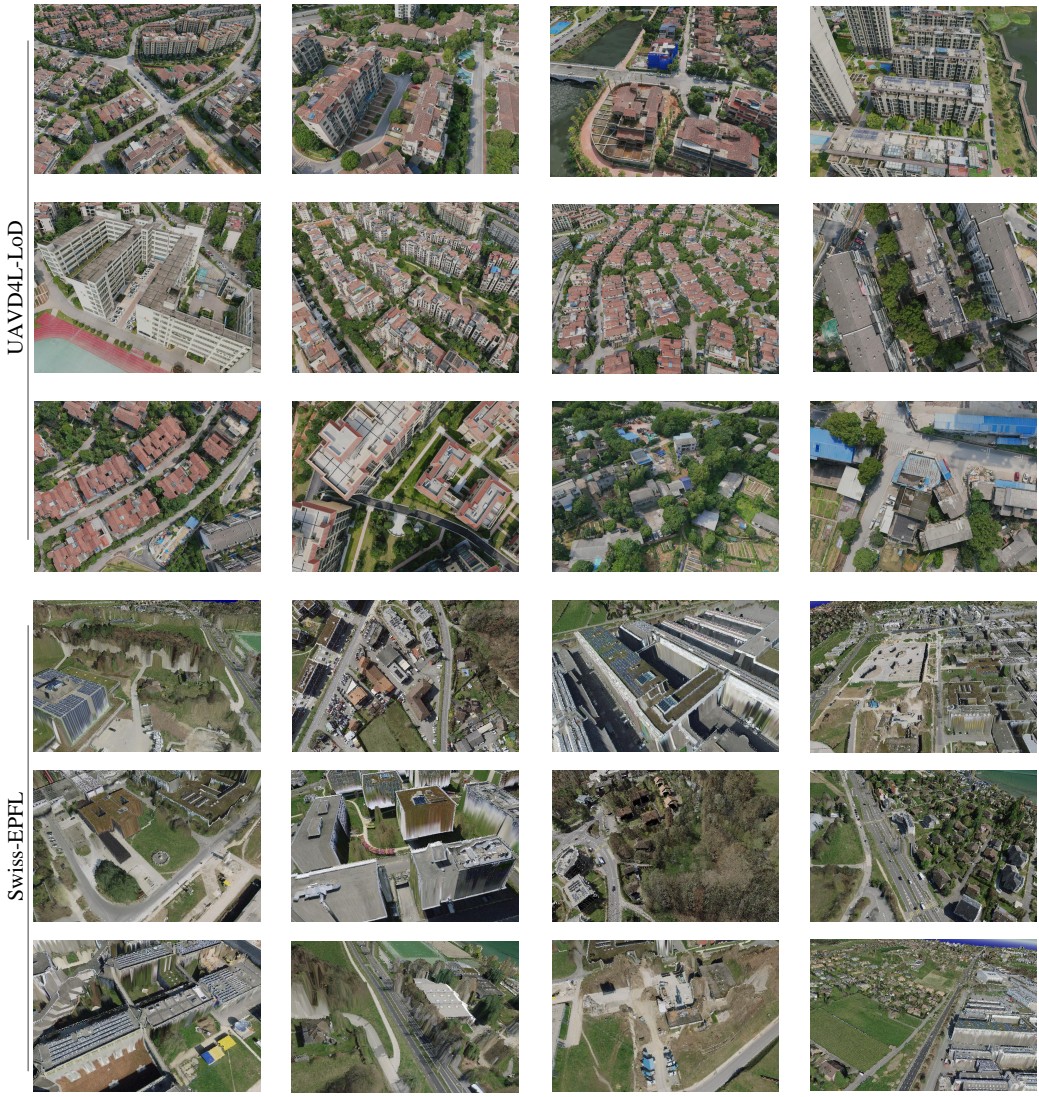

Figure 15: **Samples of the training dataset**. The UAVD4L-LoD dataset offers high-quality training set, while the Swiss-EPFL dataset suffers from lower quality, as evidenced by issues such as blurriness and voids on the sides of buildings.

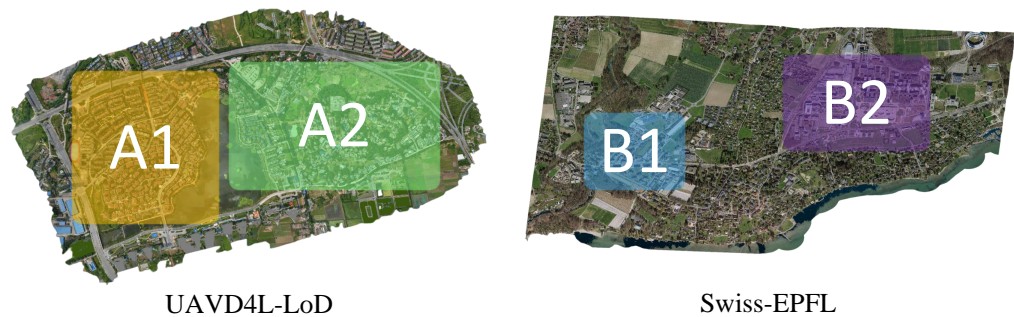

UAVD4L-LoD                                    Swiss-EPFL

Figure 16: **Region of training and testing**. We use boxes with different colors and symbols to delineate different regions.

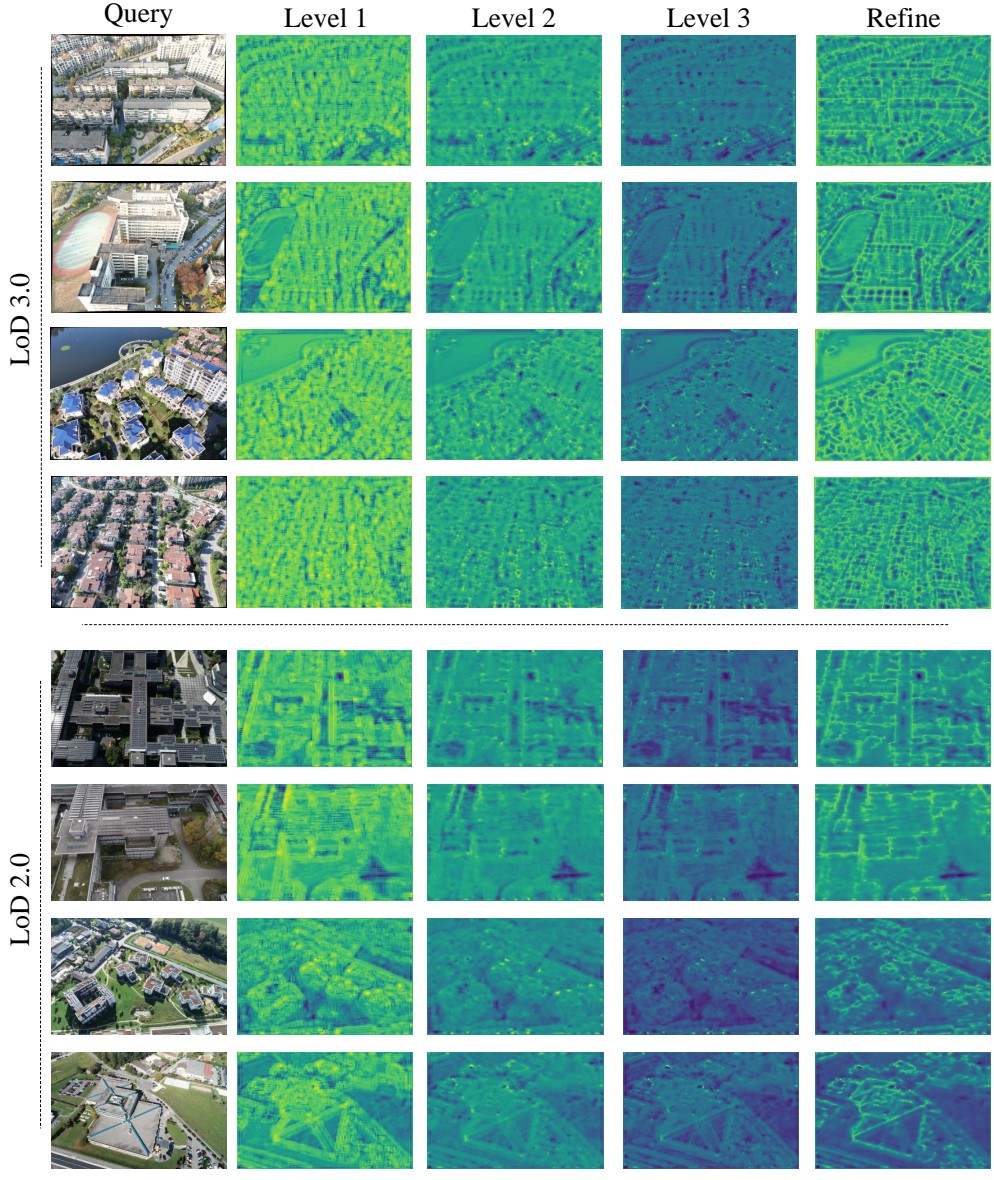

Figure 17: **Visualization of feature maps at different levels**. The feature maps at different levels reflect varying degrees of fineness in wireframe extraction.

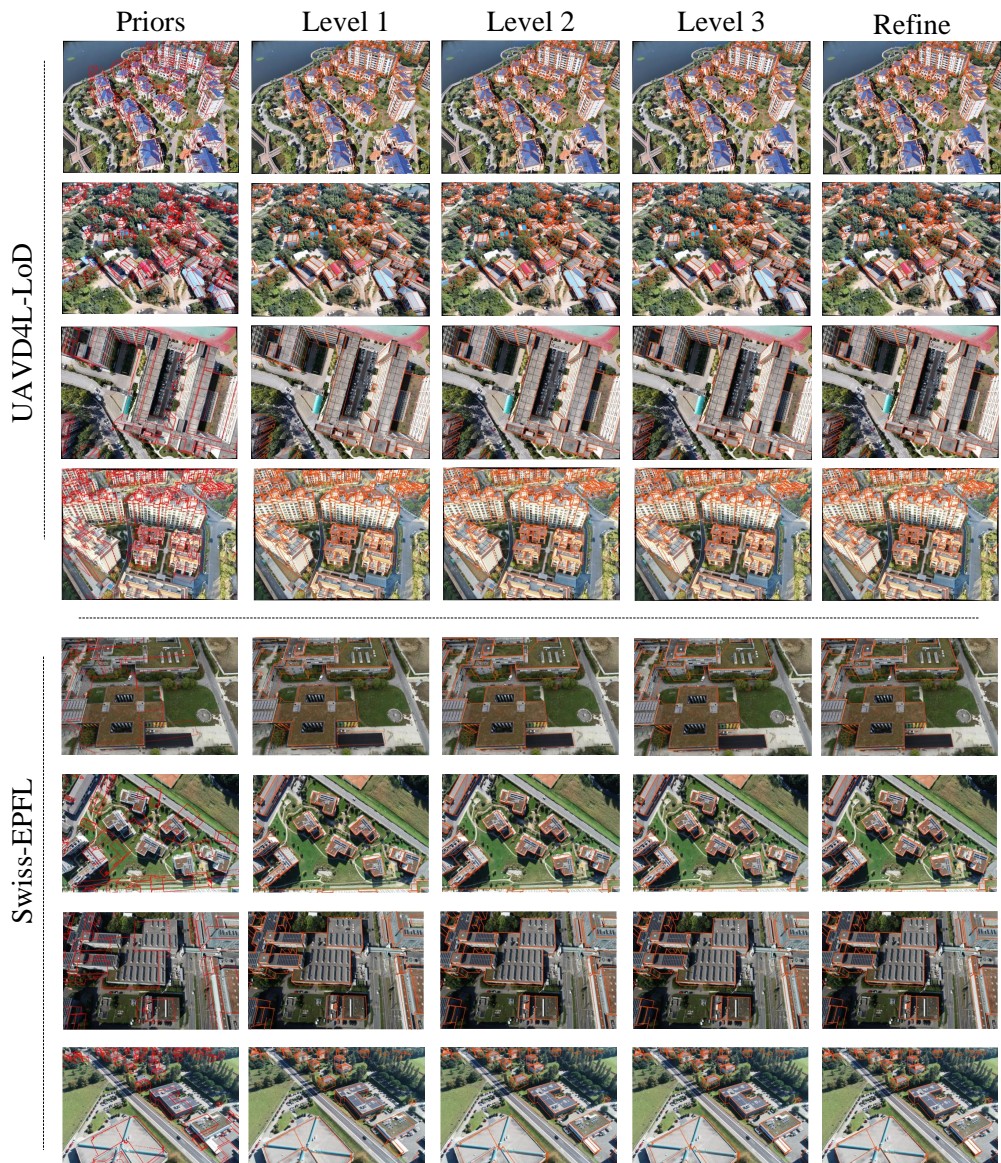

Figure 18: **Visualization of predictions at different levels**. Based on the predicted poses at each stage, we can obtain 2D projected wireframe and overlay them on the query image to check the accuracy of the poses. It can be observed that as the levels progress, the projected wireframes gradually align with the edges of the buildings. Please zoom in to see the details of the alignment.

