# OpenReview forum: "LoD-Loc: Aerial Visual Localization using LoD 3D Map with Neural Wireframe Alignment"
_NeurIPS.cc/2024/Conference — NeurIPS 2024 poster_

### Official Review · Reviewer_CDoh · 2024-06-16

**Soundness:** 2
**Presentation:** 2
**Contribution:** 2
**Rating:** 5
**Confidence:** 4

**Summary:**

This paper proposes a hierarchical scheme for pose selection to progressively compute high-quality pose in a coarse-to-fine fashion.

**Strengths:**

The scheme of the proposed method is easily understood and the paper shows good experimental results.

**Weaknesses:**

The innovation of the paper is not clearly described.

**Questions:**

This paper give a detailed description of some proposed steps, without providing the target of these steps.  As far as I am concerned, these steps are general steps for localization, what's your innovation?
The problem you want to solve and the shortcomings that current methods exist should be strengthened.
The quality of some figures can be improved.

**Limitations:**

As described above

---

> ### Author Rebuttal · Authors · 2024-08-07
>
> **Q1: “As far as I am concerned, these steps are general steps for localization, what's your innovation?”**
>
> **A1:**
> We believe our innovation could be summarized in the following parts:
>
> **Task:** We establish a novel task that utilizes Level-of-Detail (LoD) 3D maps for 6-DoF aerial visual localization, which offers advantages such as ease acquisition and maintenance, lightweight map size, and enhanced privacy preservation and policy compliance. The task has been recognized as "**practical**" by reviewers RdXx, UAbv, nkE1 and H1Dr.
>
> **Method:** We propose a novel method tailored for this task, which utilizes wireframe alignment for pose estimation. This approach is distinct from previous research that typically solves for the pose via descriptor-based matching and PnP RANSAC solver. Additionally, our method integrates several modules, such as coarse-to-fine pose cost volume reconstruction, uncertainty sampling range estimation, and differential Gauss-Newton refinement into a whole network, which can be trained in an end-to-end manner with only pose supervision. Our method has been admired by other reviewers, noted as "**novel**" from UAbv and H1Dr, "**interesting**" from RdXx and H1Dr, and "**privacy-protecting**" from H1Dr and nkE1.
>
> **Dataset:** We have release two LoD city datasets featuring LoD2.0 and LoD3.0 models, complete with RGB queries and ground-truth pose annotations. Our dataset has been recognized as "**useful**" by reviewers  RdXx, UAbv and nkE1, and is noted to “**provide interesting resources for further research and benchmarking in aerial visual localization**” by RdXx.
>
> **Result:** We have achieved "**promising**" results as noted by reviewers UAbv and H1Dr, demonstrating comparable or better performance with state-of-the-art texture-based approaches, while only using simplified LoD models to guide the localization.

---

> > ### Comment · Reviewer_CDoh · 2024-08-13
> >
> > Thanks for your response.

---

### Official Review · Reviewer_H1Dr · 2024-07-12

**Soundness:** 3
**Presentation:** 3
**Contribution:** 3
**Rating:** 8
**Confidence:** 5

**Summary:**

The paper presents a localization algorithm using LoD maps. Compared to conventional localization pipelines using SfM / SLAM maps, LoD maps are memory efficient and can offer privacy preservation. Nevertheless, due to the lack of texture in LoD maps, it is not straightforward to use pre-existing localization pipelines for LoD map localization. The paper thus presents a localization pipeline that can handle such ambiguities, by presenting a pose search / refinement pipeline based on estimating wireframe probabilities from images. Specifically, the paper suggests training CNNs to predict wireframe probabilities from images, and measures the geometric compatibility of the predicted probabilities against the LoD map. Experiments show that the proposed method can largely outperform existing CAD-based localization methods, while showing competitive performance against methods conventional methods using textured meshes and local feature descriptors.

**Strengths:**

1. The problem of localizing LoD maps seems to be a practical yet quite under-explored problem. I find the problem setup to be very interesting and can foster future works in the direction. While the performance is currently slightly below visual descriptor-based methods, I find the initial results to be promising.

2. The idea of predicting wireframe probabilities is novel and well-motivated. It seems natural to consider such structures to be extracted from 2D images to make valid comparisons against the wireframe map.

3. The writing is very clear, and the supplementary materials section is very thorough which I expect will clarify a large body of details that readers may be interested in.

**Weaknesses:**

I don't have large concerns for the paper, but below are a few suggestions:
1. Several related works seem to be missing, including some recent papers on descriptor-free visual localization:

a) Wang et al., DGC-GNN: Leveraging Geometry and Color Cues for Visual Descriptor-Free 2D-3D Matching, arXiv 2024
b) Kim et al., Fully Geometric Panoramic Localization, CVPR 2024
c) Pietrantoni et al., SegLoc: Learning Segmentation-based Representations for Privacy-Preserving Visual Localization, CVPR 2023.

2. I wonder how the proposed method could be extended for larger-scale pose search scenarios, for example if GPS / gravity direction is unknown. I feel the search space might be too large for the current grid-based search to work. Do the authors have any rough suggestions for how the work could be extended in such directions?

3. How would be proposed method compare against explicitly extracting line segments from 2D images? To be more specific, suppose one replaces the wireframe probability extraction part explained in Section 3.1 with distance transformed calculated for 2D line extractions using conventional line extractors such as LSD or DeepLSD[1]. The cost functions or features in Equations 2 and 4 could be replaced with  the distance function values, similar to line-based localization methods [2, 3]. I feel this would not fare well since there would be large number of noisy lines and lines from image textures, but still the comparisons would provide insights into why the feature extraction in Section 3.1 is necessary for good performance.

[References]
1. Pautrat et al., DeepLSD: Line Segment Detection and Refinement with Deep Image Gradients, ICCV 2023
2. Micusik et al., Descriptor free visual indoor localization with line segments, CVPR 2015
3. Kim et al., LDL: Line Distance Functions for Panoramic Localization, ICCV 2023

**Questions:**

I listed some minor questions below:
1. What are the units of delta_x, y, z in Table 11?
2. Related to Table 11, I feel it will be nice to see more experiments on the impact of the initial pose. For example, what errors would we get if we initialize from place recognition algorithms for UAVs [1]?


[References]
1. Keetha et al., AnyLoc: Towards Universal Visual Place Recognition

**Limitations:**

Yes the limitations are discussed.

---

> ### Author Rebuttal · Authors · 2024-08-06
>
> **Q1: “Several related works seem to be missing, including some recent papers on descriptor-free visual localization. […]”**
>
> **A1:** Thank you for your suggestion. We will incorporate the related works in the revised paper.
>
> **Q2: ”I wonder how the proposed method could be extended for larger-scale pose search scenarios, for example if GPS / gravity direction is unknown. […]”**
>
> **A2:** Thank you for your question. Possible solutions could include utilizing an RGB-to-LoD cross-modal retrieval network to determine the initial pose or leveraging the UAV’s sequence motion information as a prior. These strategies may effectively address the issue.
>
> Additionally, it is noteworthy that we utilize **real** built-in sensors from the UAV as priors. We believe that employing such priors to guide localization is reasonable in practical scenarios. As we know, many previous studies have used GPS and gravity direction as priors, such as [1,2,3,4,5] for GPS-based localization and [1,2,3,5,6,7,8,9,10,11] for gravity direction-based localization.
>
> **Q3: “How would be proposed method compare against explicitly extracting line segments from 2D images? […]”**
>
> **A3:** Thank you for your suggestion. We believe this is a highly valuable experiment to demonstrate the effectiveness of our method. As you suggested, we have added additional experiments named **Explicit Line Alignment**, as shown in Tables 1 and 2 of the **PDF** in the global response and the Table below.
>
> | Method                  | UAVD4L-LoD            |                  | Swiss-EPFL              |                   |
> | -----------------------| ----------------- | ---------------- | ----------------- | ----------------- |
> |                         | in-Traj            | out-of-Traj          | in-Place           | out-of-Place          |
> | Explicit Line Alignment | 10.41/16.21/24.19 | 6.93/12.64/21.62 | 11.37/21.35/33.57 | 18.99/31.39/45.91 |
> | Ours | 84.41/91.77/96.95  | 95.94/99.00/99.36 |  48.60/65.31/79.78  | 37.73/57.26/77.57   |
>
> Specifically, we extract explicit line segments using DeepLSD[12] and construct a distance field for each segment. We then replace the cost function in Equations 2 and 4 with the distance function values and solve for the pose using coarse-to-fine pose selection and Gauss-Newton refinement. The results indicate that our method surpasses **Explicit Line Alignment** by a significant margin, proving that the feature extraction in Section 3.1 is necessary for good performance. This experiment will be included in the revised paper.
>
> **Q4: “What are the units of delta_x, y, z in Table 11?”**
>
> **A4:** The units for delta_x, y, and z are **meters**. We will also include this note in caption of Table 11 in the revised paper.
>
>
> **Q5: "Related to Table 11, I feel it will be nice to see more experiments on the impact of the initial pose. For example, what errors would we get if we initialize from place recognition algorithms for UAVs [13]?"**
>
> **A5:** Thank you for your suggestion. We agree that initialization from place recognition algorithms for UAVs could be a reasonable solution for GPS-denied environments, and it warrants further experimentation. However, our method aims to localize the UAV solely using the LoD model, whereas conventional place recognition algorithms such as [13] rely on rich-information maps (i.e., satellite images), which are not available in our dataset. In future work, we aim to train a global network with the capability of RGB-to-LoD retrieval and test our algorithm with this initial pose.
>
> **Reference**
>
> [1] yan et al., Long-term Visual Localization with Mobile Sensors, CVPR 2023
>
> [2] Sarlin et al., OrienterNet: Visual Localization in 2D Public Maps with Neural Matching, CVPR 2023
>
> [3] Sarlin et al., SNAP: Self-Supervised Neural Maps for Visual Positioning and Semantic Understanding, NeurIPS2023
>
> [4] Zeisl et al., Camera pose voting for large-scale image-based localization, ICCV 2015
>
> [5] Lynen et al., Large-scale, real-time visual-inertial localization revisited, IJRR 2020
>
> [6] Albl et al., Rolling shutter absolute pose problem with known vertical direction, CVPR  2016
>
> [7] Kukelova et al., Closed- form solutions to minimal absolute pose problems with known vertical direction, ACCV 2010
>
> [8] Svärm et al., City-scale localization for cameras with known vertical direction, T-PAMI 2016
>
> [9] Sweeney et al., Efficient computation of absolute pose for gravity-aware augmented reality , ISMAR  2015
>
> [10] Fragoso et al., gdls*: Gen- eralized pose-and-scale estimation given scale and gravity priors, CVPR 2020
>
> [11] Sweeney et al., gdls: A scalable solution to the generalized pose and scale problem, ECCV  2014
>
> [12] Pautrat et al., DeepLSD: Line Segment Detection and Refinement with Deep Image Gradients, ICCV 2023
>
> [13] Keetha et al., AnyLoc: Towards Universal Visual Place Recognition, IEEE Robotics and Automation Letters 2023

---

> > ### Comment · Reviewer_H1Dr · 2024-08-13
> >
> > Thank you for the detailed response: it has answered most of my questions. Therefore I will raise my score.

---

### Official Review · Reviewer_nkE1 · 2024-07-12

**Soundness:** 2
**Presentation:** 2
**Contribution:** 2
**Rating:** 4
**Confidence:** 4

**Summary:**

This paper introduces a method for visual localization of Unmanned Aerial Vehicles (UAVs) using a Level-of-Detail (LoD) 3D map and neural wireframe alignment.  The UAV sensor provides coarse pose estimation. Then, LoD-Loc hierarchically builds a cost volume for uniformly-sampled pose hypotheses to describe pose probability distribution.  A pose with maximum probability is selected as the correct one. Each cost within this volume measures the degree of line alignment  between projected and predicted wireframes. It also refines the previous result with a differentiable Gaussian-Newton method. The paper presents an approach to aerial visual localization with a strong emphasis on practical applicability and privacy considerations.

**Strengths:**

1. This work gives an approach to aerial visual localization that does not require complex 3D representations.

2. The method uses LoD 3D maps, which are easier to acquire and maintain, have a smaller footprint.

3. it contributes new datasets for the research community to use for training and evaluation.

**Weaknesses:**

1. The importance of the major contribution of "Level-of-Detail" is not clearly described. Acctually, Level-of-Detail has been widely investigated.

2. The paper does not explicitly mention the computational efficiency or resource requirements, which could be a concern for real-time applications on UAVs with limited processing power.

3. The reliance on a coarse pose provided by the UAV sensor as an initial input may affect the method's robustness in scenarios where the initial pose is significantly off.

**Questions:**

Is it possible that the method's performance might degrade in environments with poor GPS signal or where the UAV is operating in areas without prior map knowledge?

**Limitations:**

1. The method assumes the availability of a gravity direction and location prior, which might not always be accurate or available, especially in GPS-denied environments.

2. The generalizability of the method to other types of UAVs or different environmental conditions is not fully explored in the paper.

3. The paper does not discuss the potential impact of dynamic elements in the environment (such as moving vehicles or people) on pose estimation accuracy.

4. The robustness of the method against various weather conditions or times of day is not addressed.

---

> ### Author Rebuttal · Authors · 2024-08-06
>
> **Q1: “The importance of the major contribution of "Level-of-Detail" is not clearly described. Actually, Level-of-Detail has been widely investigated.”**
>
> **A1:** The LoD model is a well-established concept used in building reconstruction and design to manage the complexity of 3D models. Compared to Structure-from-Motion (SfM) and textured mesh models, LoD models offer several advantages, including ease of acquisition and maintenance, lightweight map size, and enhanced privacy preservation and policy compliance. While we acknowledge that LoD has been extensively investigated in these fields, previous research has not leveraged this model to guide 6-DoF visual localization. One of the key contributions of our paper is the first use of LoD 3D maps for 6-DoF aerial visual localization.
>
> **Q2: “The paper does not explicitly mention the computational efficiency or resource requirements.”**
>
> **A2:** Please refer to Table 3 in the **PDF** of the global response.
>
> **Q3: “The reliance on a coarse pose provided by the UAV sensor […] Is it possible that the method's performance might degrade in environments with poor GPS signal or where the UAV is operating in areas without prior map knowledge?”**
>
> **A3:**  As indicated in “Table 11: Impact of the initial pose for LoD-Loc,” our method’s performance indeed deteriorates with declining GPS quality. This phenomenon is also discussed in the Limitations section of our paper.
>
> However, it is noteworthy that we utilize **real** built-in sensors from the UAV as priors. We believe that employing such priors to guide localization is reasonable in practical scenarios. Additionally, many previous studies have also used GPS and gravity direction as priors, such as [1,2,3,4,5] for GPS-aided localization and [1,2,3,5,6,7,8,9,10,11] for gravity direction-aided localization.
>
> In cases where GPS is denied, we propose that employing an RGB-to-LoD cross-modal retrieval or utilizing the UAV’s sequence motion information to find the initial position could effectively address this issue.
>
> Similar to [12,13,14], our method relies on 3D maps for localization. Without a map, it is impossible to determine the UAV’s absolute pose; only the relative motion of the UAV sequence can be known.
>
> **Q4: “The method assumes the availability of a gravity direction and location prior, which might not always be accurate or available, especially in GPS-denied environments.”**
>
> **A4:** Please refer to **A3**.
>
> **Q5: “The generalizability of the method to other types of UAVs or different environmental conditions is not fully explored in the paper.”**
>
> **A5:** In fact, we conducted localization using various types of UAVs across different environments. In the UAVD4L-LoD dataset, we captured scenarios labeled as “in-traj.” and “out-of-traj.” as detailed in the table below:
>
> | **Name** | **Capture device**  | **Capture pitch angle**   |  **Capture height**   |   **Capture route**  |
> |:--------:|:--------:|:----------:|:----------:|----------|
> | in-Traj.   | DJI M300+H20t    |  0&deg; or 45&deg;  |  120m   |  Zig-zag flight on a selected region |
> | out-of-Traj.   | DJI Mavic3 Pro  | 30&deg; ~ 60&deg;     | 90m ~ 150m   |  Manually controlled flight on the map   |
>
> In the EPFL-Swiss dataset, the DJI Phantom 4 RTK is used to collect data from both “in-Place” and “out-of-Place” regions. For more details, please refer to [15].
>
> **Q6: “The paper does not discuss the potential impact of dynamic elements in the environment (such as moving vehicles or people) on pose estimation accuracy.”**
>
> **A6:** Theoretically, our method uses the building's wireframe as a cue for localization, thus dynamic elements in the environment have little impact on our algorithm. We will incorporate a discussion on this aspect in the revised paper.
>
> **Q7: “The robustness of the method against various weather conditions or times of day is not addressed.”**
>
> **A7:** As demonstrated in the **video demo** in the **Supplementary Material**, our method is capable of performing localization at night using thermal cameras. Besides, we are in the process of collecting more data to validate the robustness of our method, leveraging the scalable solution we provide for capturing query data and ground truth annotations. More data under various weather conditions or times of day will be incorporated in our project to further present the robustness of our method.
>
> **Reference**
>
> [1] yan et al., Long-term Visual Localization with Mobile Sensors, CVPR 2023
>
> [2] Sarlin et al., OrienterNet: Visual Localization in 2D Public Maps with Neural Matching, CVPR 2023
>
> [3] Sarlin et al., SNAP: Self-Supervised Neural Maps for Visual Positioning and Semantic Understanding, NeurIPS 2023
>
> [4] Zeisl et al., Camera pose voting for large-scale image-based localization, ICCV 2015
>
> [5] Lynen et al., Large-scale, real-time visual-inertial localization revisited, IJRR 2020
>
> [6] Albl et al., Rolling shutter absolute pose problem with known vertical direction, CVPR  2016
>
> [7] Kukelova et al., Closed- form solutions to minimal absolute pose problems with known vertical direction, ACCV 2010
>
> [8] Svärm et al., City-scale localization for cameras with known vertical direction, T-PAMI 2016
>
> [9] Sweeney et al., Efficient computation of absolute pose for gravity-aware augmented reality , ISMAR  2015
>
> [10] Fragoso et al., gdls*: Gen- eralized pose-and-scale estimation given scale and gravity priors, CVPR 2020
>
> [11] Sweeney et al., gdls: A scalable solution to the generalized pose and scale problem, ECCV  2014
>
> [12] Sattler et al., Benchmarking 6dof outdoor visual localization in changing conditions, CVPR 2018
>
> [13] Sarlin et al., From Coarse to Fine: Robust Hierarchical Localization at Large Scale, CVPR 2019
>
> [14] Miao et al., A Survey on Monocular Re-Localization: From the Perspective of Scene Map Representation, IEEE Transactions on Intelligent Vehicles, 2024
>
> [15]Yan et al., Crossloc: Scalable aerial localization assisted by multimodal synthetic data, CVPR 2024

---

> ### Author Response · Authors · 2024-08-13
>
> Dear Reviewer nkE1,
>
> Thank you for your review. May I politely ask if you have any other questions regarding this work? If any further clarification is needed, please do not hesitate to reach out. Your comments is important to this work.
>
> Best regards,
>
> Authors

---

### Official Review · Reviewer_UAbv · 2024-07-13

**Soundness:** 3
**Presentation:** 3
**Contribution:** 3
**Rating:** 6
**Confidence:** 4

**Summary:**

The paper presents an approach to provide an accurate camera pose of an aerial image according to a LoD 3D map, which is a lightweight representation to store the key information of a known map. For each query image, multi-scale wireframe probability maps are extracted by a trained U-net. These maps are then used to select sampled poses from cost volume. Such process is executed in a coarse-to-fine manner. Lastly, Gauss-Newton is used to refine the final pose by minimizing the projection error of 3D wireframe points.

Two LoD city datasets are also released with the paper.

**Strengths:**

The proposed method is able to estimate an accurate pose from a LoD model and the experiments show the method can achieve generally comparable results compared to the baseline methods, among which one uses a high-cost texture mesh map for localization.

The design of coarse-to-fine pose selection is novel in this task and is validated to be useful in the experiments section.

The release of the LoD dataset could be beneficial to the aerial localization research field.

**Weaknesses:**

In this paper, the definition of "in-Traj" and "out-of-Traj" is not clearly provided. I guess this may mean the data used to train the model and those in the test stage. Please clarify these concepts in the paper.

The proposed method is significantly worse than UAVD4L in "out-of-place" part of Swiss-EPFL dataset. The authors explain that this is due to the relatively low quality of the dataset and LoD2.0 model. However, this shows that the method is much less robust to the data quality and it only works when high-quality data is provided, which can be very costly.

**Questions:**

Do you have any process to deal with the sky pixels in query images?

**Limitations:**

The authors have mentioned the limitations of the method and the potential privacy impacts.

---

> ### Author Rebuttal · Authors · 2024-08-06
>
> **Q1: “The definition of "in-Traj" and "out-of-Traj" is not clearly provided.”**
>
> **A1:** Sorry for the confusion. The 'in-Traj' and 'out-of-Traj' scenarios refer to two distinct query image collections as introduced in  **A.2 Query Image Collection** in the **Appendix** and depicted in "Figure 6: Flight trajectories of query images in the UAVD4L-LoD dataset." The main differences are summarized in the following table:
>
> | **Name** | **Capture device**  | **Capture pitch angle**   |  **Capture height**   |   **Capture route**  |
> |:--------:|:--------:|:----------:|:----------:|----------|
> | in-Traj.   | DJI M300+H20t    |  0&deg; or 45&deg;  |  120m   |  Zig-zag flight on a selected region |
> | out-of-Traj.   | DJI Mavic3 Pro  | 30&deg; ~ 60&deg;     | 90m ~ 150m   |  Manually controlled flight on the map   |
>
> In the revised version of the paper, we will provide clear definitions of “in-Traj.” and “out-of-Traj.,” along with a detailed explanation of the key distinctions between these scenarios.
>
> **Q2: “The proposed method is significantly worse than UAVD4L in "out-of-place" part of Swiss-EPFL dataset […]”**
>
> **A2:** Our work aims to explore the use of LoD maps for 6-DoF localization in the air.  We have designed a method specifically tailored for this task and collected two datasets featuring LoD3.0 and LoD2.0 models to validate our algorithm and benefit future research. The experimental results demonstrate that our method exhibits promising performance compared to visual descriptor-based methods.
>
> Regarding the poorer performance in the ‘out-of-place’ part of the Swiss-EPFL dataset, we identify several contributing factors:
>
> First, the image quality used to train our network in the Swiss-EPFL dataset is relatively low, as shown in “Figure 14: Samples of training dataset” in the **Appendix**. We observe that model trained on UAVD4L dataset (even with LoD3.0) using high-quality image, performs better than model trained directly on Swiss-EPFL dataset, as indicated in the last two rows of "Table 12: Cross-scene generalization" in the **Appendix**.
>
> Second, we find that using LoD3.0 maps results in better localization accuracy compared to LoD2.0. This is intuitive because our method relies on wireframe representations, and the richer lines in LoD3.0 maps lead to improved results. These results give us insight that for LoD2.0 or lower LoD models, regions could be used as cues for localization in future work.
>
> Third, it is noteworthy that our results based on the LoD2.0 model are still quite good. In fact, in the "in-place" part, our method even outperforms SOTA textured-based methods.
>
> **Q3: “Do you have any process to deal with the sky pixels in query images?”**
>
> **A3:** Our method employs pose as supervision and trains the network to extract image features in an end-to-end manner. We do not specifically process the sky pixels in the query images.

---

> ### Comment · Reviewer_UAbv · 2024-08-13
>
> I appreciate the authors' response. It addresses my questions. I will raise my rating.

---

> > ### Author Response · Authors · 2024-08-13
> >
> > Thanks for your time. May I humbly inquire about any specific actions you would recommend that could potentially increase the score as per your initial review? Because, we notice that there has been no change on the rating score in the system. If there are any additional clarifications needed, please do not hesitate to reach out.

---

> ### Author Response · Authors · 2024-08-14
>
> Dear Reviewer UAbv,
>
> We hope this message finds you well.
>
> We appreciate your responses and are pleased to have addressed your concerns. We noticed that the system not yet updated your rating. We are unsure if this is due to a display issue with the system or if you have any other concerns regarding this work. If there are any additional clarifications needed, please do not hesitate to reach out.
>
>
> Best regards,
>
> Authors

---

> > ### Comment · Reviewer_UAbv · 2024-08-14
> >
> > Updated

---

### Official Review · Reviewer_RdXx · 2024-07-13

**Soundness:** 3
**Presentation:** 3
**Contribution:** 3
**Rating:** 5
**Confidence:** 4

**Summary:**

The paper proposes a method named LoD-Loc for visual localization of unmanned aerial vehicles (UAVs) using Level-of-Detail (LoD) 3D maps. Unlike existing algorithms that rely on complex 3D representations, LoD-Loc aligns wireframes derived from LoD model projections with the wireframes predicted by a CNN Unet to estimate the UAV's pose. To solve the problem, given a coarse pose provided by the sensor, this paper utilizes a hierarchical cost volume to pose hypothesis evolution. And eventually, a differentiable Gaussian-Newton algorithm for 6-DoF pose refinement.  This work also contributes two datasets to facilitate further research in this area and provides a comprehensive evaluation of the proposed method's performance. The experimental results demonstrate that LoD-Loc outperforms current state-of-the-art methods using textured 3D models, offering advantages in ease of acquisition, lightweight map size, and privacy preservation.

**Strengths:**

The strengths of the paper are the following:

**Interesting Methodology:**

The proposed LoD-Loc method introduces an interesting approach to visual localization using Level-of-Detail (LoD) 3D maps, which is less reliant on complex 3D representations. This makes the method more accessible and scalable compared to current textured 3D model-based approaches.

**Application in practical scenarios:**

LoD maps are easier to acquire and maintain and have a smaller file size. These advantages make LoD-Loc more feasible for real-world applications, particularly in large-scale environments where data storage efficiency is a critical requirement.

**Contribution of Datasets:**

The paper contributes two largest-scale air localization datasets (LoD3.0 and LoD2.0) with RGB queries and ground-truth pose annotations. These datasets provide interesting resources for further research and benchmarking in aerial visual localization.

**Weaknesses:**

**Experiment Result**

The author mentioned CadLoc "cannot even localize the query image at a coarse threshold at $5m -5^{\circ}$". However, what I observed from Table 2 even Table 3 is the result for CadLoc are all zeros. Is that a mistake?

In Table 2, the authors show their method surpasses the SOTA textured 3D-model-based method in the $2m -5^{\circ}$ and $3m -5^{\circ}$ and "out-of-Traj."
However, the author does not explain why their method can perform better than textured 3D-model-based methods in "out-of-Traj" scenarios but fails in "out-of-Traj" scenarios.

In the Appendix, the author mentioned that "in-Traj." and "out-of-Traj." represent trajectory-based and free-flight scenarios, respectively.
But I didn't catch what might be the key difference between these two scenarios. Does that mean these methods are actually verified in different trajectories in the "out-of-Traj." scenarios?
If so, I think it is not a fair comparison. If not, I didn't get the key difference between the "in-Traj." and "out-of-Traj." scenarios. It is suggested to explain why the proposed method performs better than the SOTA textured 3D-model-based method considering they use high-precision texture 3D information.

While the author mentions they conducted experiments on many other image retrieval-and-matching and textured 3D models-based approaches, only the results of UAVD4L and CadLoc are presented. It would be better to compare with more methods in each type to show a more comprehensive result.

**Computational Cost**

Although not mentioned, the methods involving hierarchical cost volume construction and differentiable optimization algorithms can be computationally intensive. This could pose challenges for real-time applications, especially on resource-constrained UAVs. For more comprehensive research, it would be better to provide a computational cost comparison for that.

**Presentation**

I think the writing and presentation skills in this paper could be further improved. Such as the presentation of Table 1 and the Contribution sections. Besides, I would suggest the author double-check their result in the presented Tables. Besides, I would suggest that the author add more explanations for their experiment result instead of just simply displaying them.

**Questions:**

The proposed idea in general sounds okay to me. However, my questions are mainly about the result of the experiment result listed in the Weaknesses. I think the result present is questionable, insufficient, and lacks enough explanation to support the proposed method.
Besides, I think the writing and presentation need to be further improved for this paper to be accepted.

**Limitations:**

Yes

---

> ### Author Rebuttal · Authors · 2024-08-06
>
> **Q1.1: "What I observed from Table 2 even Table 3 is the result for CadLoc are all zeros. Is that a mistake?"**
>
> **A1.1:** Upon thorough verification, we confirm that the results for CadLoc in both Table 2 and Table 3 are indeed zeros. The primary reason may be that advanced RGB-based retrieval and matching algorithms are not effective for cross-modality tasks involving RGB-to-LoD images. For further insights, please refer to "Figure 12: Failure retrieval cases of baselines” and “Figure 13: Failure matching cases of baselines” in the **Appendix**, which illustrate the limitations of these methods.
>
> **Q1.2: "The author does not explain why their method can perform better than textured 3D-model-based methods in "out-of-Traj" scenarios but fails in "out-of-Traj" scenarios."**
>
> **A1.2:** We acknowledge that reviewer would like to figure out why our method performs better than textured 3D-model-based methods in "out-of-Traj." but lags behind in "in-Traj." scenarios. We apologize for the lack of detailed explanations and will incorporate additional clarifications in the revised paper.
>
> Firstly, we provide a comprehensive comparison of 'in-Traj.' and 'out-of-Traj.' scenarios. Specifically, 'in-Traj' and 'out-of-Traj' scenarios refer to two distinct query image collections as introduced in the **Appendix A2** and depicted in "Figure 6: Flight trajectories of query images in the UAVD4L-LoD dataset." The main differences are summarized in the following table:
>
> | **Name** | **Capture device**  | **Capture pitch angle**   |  **Capture height**   |   **Capture route**  |
> |:--------:|:--------:|:----------:|:----------:|----------|
> | in-Traj.   | DJI M300+H20t    |  0&deg; or 45&deg;  |  120m   |  Zig-zag flight on a selected region |
> | out-of-Traj.   | DJI Mavic3 Pro  | 30&deg; ~ 60&deg;     | 90m ~ 150m   |  Manually controlled flight on the map   |
>
> Secondly, upon careful examination of the failure cases in the 'in-Traj.' scenario, we found that the preset zig-zag route on a selected region resulted in some images capturing only the corners of houses, as shown in Figure 1 in the **PDF** of the global response. This led to poorer performance in strict 2m-2° metrics. However, it is important to note that for coarse metrics in the ‘in-Traj’ scenario, our method achieves comparable or superior results. For instance, we achieve 96.95% on 5m-5° while the closest baseline achieves 92.14%.
>
> **Q1.3: "I didn't catch what might be the key difference between these two scenarios ( 'in-Traj.' and 'out-of-Traj.') ..."**
>
> **A1.3:** Please refer to **A1.2** for the key difference between 'in-Traj.' and 'out-of-Traj.' scenarios.
>
> **Q1.4: "It is suggested to explain why the proposed method performs better than the SOTA textured 3D-model-based method considering they use high-precision texture 3D information."**
>
> **A1.4:** First, compared to the SOTA textured-based approach, our method employs distinct cues for localization. The textured-based method determines pose by optimizing the re-projection error of corresponding 2d-3d points. Conversely, our novel approach aligns the 3D wireframe projection to solve the pose. Given that building structures are primarily composed of wireframes, we posit that leveraging such cues for localization is a well-motivated solution.
>
> Second, 3D-model-based methods typically employ a two-stage scheme, which involves building 2d-3d matches and then solving the pose with PnP RANSAC. Our method directly solves for the pose in an end-to-end manner, which may result in better pose accuracy.
>
> Third, we introduce several important modules to improve performance, such as coarse-to-fine pose cost volume reconstruction, uncertainty sampling range estimation, and differential Gauss-Newton refinement. These factors contribute to the superior performance of our method.
>
> We will provide additional explanations for the experimental results in the revised paper.
>
> **Q1.5: "It would be better to compare with more methods in each type (UAVD4L and CadLoc) to show a more comprehensive result."**
>
> **A1.5:** The UAVD4L and CadLoc frameworks, which are the SOTA and the most suitable for this task, are designed for texture models and CAD models, respectively.  To provide more comparison, we include SOTA matching methods (RoMA[1] and e-Loftr[2]) within the UAVD4L and CadLoc frameworks. Please refer to Table 1 and Table 2 in the **PDF** of the global response for detailed comparisons.
>
> **Q2: "For more comprehensive research, it would be better to provide a computational cost comparison for that."**
>
> **A2:** Please refer to Table 3 in the **PDF** of the global response. We conduct test experiments on a single batch (Batch Size = 1) of images using the NVIDIA GeForce RTX 4090 device, and list the average peak CUDA usage as well as the average inference time.
>
> **Q3.1: "I think the writing and presentation skills in this paper could be further improved. Such as the presentation of Table 1 and the Contribution sections."**
>
> **A3.1:** Thanks a lot for your suggestion. We have updated our manuscript to enhance the presentation of Table 1 and the Contribution sections, providing more detailed and clear information.
>
> **Q3.2: "I would suggest the author double-check their result in the presented Tables."**
>
> **A3.2:** Thanks a lot for your suggestion. Please refer to **A1.1** for the verification of the results presented in the Tables.
>
> **Q3.3: "I would suggest that the author add more explanations for their experiment result instead of just simply displaying them."**
>
> **A3.3:** Thanks a lot for your suggestion. We have added more explanations, including why our method surpasses 3D-model-based methods and why our method performs better in “out-of-Traj.” scenarios compared to “in-Traj.” scenarios.
>
> **Reference**
>
> [1] Edstedt et al., RoMa: Robust Dense Feature Matching, CVPR 2024
>
> [2] Wang et al., Efficient LoFTR: Semi-Dense Local Feature Matching with Sparse-Like Speed, CVPR 2024

---

> > ### Comment · Reviewer_RdXx · 2024-08-13
> >
> > Thanks for the detailed response from the author. I think it addresses my concern and provides a better presentation. I will raise my rating.

---

> > > ### Author Response · Authors · 2024-08-13
> > >
> > > Thank you for your time. May I humbly inquire about any specific actions you would recommend that could potentially increase the score as per your initial review? Because, we notice that there has been no change on the rating score in the system. If there are any additional clarifications needed, please do not hesitate to reach out.

---

### Author Rebuttal · Authors · 2024-08-06

We extend our sincere thanks to all reviewers for their insightful feedback. We are honored that our work has been recognized as "**practical**" (RdXx, UAbv, nkE1, H1Dr), our dataset as "**useful**" (RdXx, UAbv, nkE1), our method as "**novel**" (UAbv, H1Dr), "**interesting**" (RdXx, H1Dr), and "**privacy-protecting**" (H1Dr, nkE1). Additionally, our results have been described as "**promising**" (UAbv, H1Dr, CDoh), and our writing as "**clear**" (H1Dr, CDoh). We hope that our work indeed "**provides interesting resources for further research in aerial visual localization**" (RdXx) and "**can foster future works in this direction**" (H1Dr).

We express our gratitude to the reviewers for acknowledging our work, which is the first effort to utilize LoD 3D maps for  aerial 6-DoF localization. Inspired by your helpful comments, we incorporated the following changes into the next/final revision of our paper:

- **Additional Baseline Experiments:** We conducted further baseline experiments using RoMA[1] and e-Loftr[2] within the CadLoc and UAVD4L frameworks. Explicit line segment localization experiments have also been included for comparative analysis (refer to Table 1 and 2 in the attached PDF).
- **Computational Cost Comparison:** We added a comprehensive comparison of computational costs and resource requirements to offer more comprehensive research (refer to Table 3 in the attached PDF).
- **Clarification of Terms:** We provided the definition of "in-Traj." and "out-of-Traj." along with a detailed explanation of the key differences between them (see Table 4 of the attaches PDF).
- **Enhanced Explanations:** We updated our manuscript to include more explanations for their experiment results, moving beyond mere presentation.
- **Manuscript Corrections:** We updated our manuscript to fix minor mistakes and reduce the risk of misunderstandings.

Reference：

[1] Edstedt et al., RoMa: Robust Dense Feature Matching, CVPR 2024

[2] Wang et al., Efficient LoFTR: Semi-Dense Local Feature Matching with Sparse-Like Speed, CVPR 2024

---

### Comment · Area_Chair_RaKU · 2024-08-12
**To all reviewers**

Dear reviewers,

This paper received mixed ratings, the author(s) have provided a rebuttal. It would be appreciated if reviewers could carefully read the rebuttal and let the authors and the AC know if it has changed your mind (and detailed justification). Your opinion will play a significant role in the final decision of this paper. Thank you so much.

Best Regards,

AC

---

### Decision · Program_Chairs · 2024-09-25

**Decision:**

Accept (poster)

**Comment:**

This paper received a mixed recommendation (54568), where a majority of the reviewers voted for acceptance.
The major concern from the only negative reviewers is related to experimental settings, the authors provided a rebuttal and justifications, while this reviewer has no response. The AC has checked the comments and paper carefully and found that the comments of the positive reviewers are more convincing, thus recommending an acceptance.